# Multispecies-coadsorption-induced rapid preparation of graphene glass fiber fabric and applications in flexible pressure sensor

Kun Wang [1,7], Xiucai Sun[1,2,7], Shuting Cheng[2,3,7], Yi Cheng [1], Kewen Huang [1], Ruojuan Liu[1,2], Hao Yuan [1,2], Wenjuan Li[1,2], Fushun Liang[1,2], Yuyao Yang[1,2], Fan Yang[1,2], Kangyi Zheng[2,4], Zhiwei Liang[2,5], Ce Tu [2], Mengxiong Liu[1,2], Mingyang Ma [1,2], Yunsong Ge[1,2], Muqiang Jian [1,2,6], Wanjian Yin [2,4], Yue Qi [2] ✉ & Zhongfan Liu [1,2] ✉

Direct chemical vapor deposition (CVD) growth of graphene on dielectric/insulating materials is a promising strategy for subsequent transfer-free applications of graphene. However, graphene growth on noncatalytic substrates is faced with thorny issues, especially the limited growth rate, which severely hinders mass production and practical applications. Herein, graphene glass fiber fabric (GGFF) is developed by graphene CVD growth on glass fiber fabric. Dichloromethane is applied as a carbon precursor to accelerate graphene growth, which has a low decomposition energy barrier, and more importantly, the produced high-electronegativity Cl radical can enhance adsorption of active carbon species by $Cl–CH_2$ coadsorption and facilitate H detachment from graphene edges. Consequently, the growth rate is increased by ~3 orders of magnitude and carbon utilization by ~960-fold, compared with conventional methane precursor. The advantageous hierarchical conductive configuration of lightweight, flexible GGFF makes it an ultrasensitive pressure sensor for human motion and physiological monitoring, such as pulse and vocal signals.

Graphene possesses a range of excellent physical properties, such as high electrical and thermal conductivities[1–3], making it a promising candidate for wide applications in electronic devices, transparent electrodes, heat dissipation modules, etc.[4–8]. The scalable fabrication of high-quality graphene is the foremost step for its industrial applications. Since the first report of graphene chemical vapor deposition (CVD) growth on Cu foil by the Ruoff group in 2009[9], graphene CVD growth on catalytic metallic substrates has been regarded as one of the most promising strategies for the mass production of high-quality graphene. Significant developments have been achieved afterward, such as the preparation of single-crystal or superclean graphene[10,11], as well as the large-scale graphene films and 4–8 inch graphene wafers[5,12]. However, for further applications, the graphene films grown on metal substrates always need to be transferred onto the target application substrates (usually dielectrics or insulators) through a complicated, time-consuming, and cost-ineffective process[13]. In addition, wrinkles,

[1]Centre for Nanochemistry, Beijing Science and Engineering Centre for Nanocarbons, Beijing National Laboratory for Molecular Sciences, College of Chemistry and Molecular Engineering, Peking University, Beijing, China. [2]Beijing Graphene Institute (BGI), Beijing, China. [3]State Key Laboratory of Heavy Oil Processing, College of Science, China University of Petroleum, Beijing, China. [4]College of Energy, Soochow Institute for Energy and Materials Innovations (SIEMIS), Jiangsu Provincial Key Laboratory for Advanced Carbon Materials and Wearable Energy Technologies, Soochow University, Suzhou, China. [5]Guangdong Provincial Key Laboratory of Quantum Engineering and Quantum Materials, School of Physics, South China Normal University, Guangzhou, China. [6]Key Laboratory of Organic Optoelectronics and Molecular Engineering of the Ministry of Education, Department of Chemistry, Tsinghua University, Beijing, China. [7]These authors contributed equally: Kun Wang, Xiucai Sun, Shuting Cheng. ✉e-mail: qiyue@bgi-graphene.com; zfliu@pku.edu.cn

cracks, and contamination are inevitably introduced into graphene during the transferring process, which will severely degrade the excellent intrinsic properties of graphene[14–16].

Given the above situation, the direct CVD growth of graphene on dielectrics or insulators (SiO$_2$, Al$_2$O$_3$, Si$_3$N$_4$, etc.) to obtain the new advanced graphene composite materials for the subsequent transfer-free applications is becoming increasingly imperative[17–19]. Nevertheless, graphene CVD growth on these substrates is usually considerably time-consuming (hours to days) due to the lack of catalytic capability of the nonmetallic substrates[20–22]. Consequently, the mass production of these graphene composite materials is largely limited, and high energy consumption is caused considering the high CVD growth temperature (>1000 °C) of graphene.

Up to now, many efforts have been devoted to improving the CVD growth rate of graphene on noncatalytic substrates. External metal catalysts (such as Cu[18], Ni[23], and Ga[24]) were introduced into the CVD system to catalyze the decomposition of carbon precursors and graphene growth. However, the poor uniformity and metal residues largely limited the industrial applications of the as-grown graphene. Moreover, carbon precursors with low decomposition energy barriers (such as ethanol[25], acetylene[26], and cyclohexane[27]) were applied to replace the conventional methane precursor, which could accelerate graphene growth by producing richer active carbon species during their decomposition. Notably, graphene CVD growth consists of several elementary steps, i.e., decomposition of carbon precursors, adsorption and nucleation of active carbon species, and further growing coalescence of graphene domains[28]. Accordingly, the reasonable modulations for each of these elementary steps will hopefully accelerate graphene growth. Therefore, there is still huge room for the improvement of graphene CVD growth rate on the noncatalytic substrates.

Glass fiber is a commercial lightweight structural material with outstanding mechanical flexibility and strength, high-temperature resistance, as a unique macrostructure, which is thus a promising CVD growing substrate of graphene, and also a valuable carrier for the atomic graphene layers during the subsequent transfer-free applications[29–32]. Herein, graphene glass fiber fabric (GGFF) was developed by CVD growing graphene on glass fiber fabric (GFF, ~99.9% SiO$_2$). Importantly, to overcome the growth rate limit for graphene on the noncatalytic GFF, dichloromethane, a widely used organic solvent in industry, was used as the carbon precursor for graphene CVD growth to increase the growth rate through accelerating the multiple CVD elementary steps, instead of one single elementary step as commonly reported in other precursor-modulating systems[33]. The dichloromethane precursor appears to be more prone to pyrolysis in the gas phase than the conventional methane precursor, and more importantly, the produced Cl radical can enhance the adsorption of active carbon species by the interesting Cl–CH$_2$ coadsorption on GFF substrate. Actually, the multiple adsorbates coadsorption, such as NO + NH$_3$, CO + C$_2$H$_4$, and CO + K, have been widely applied for promoting catalytic reactions and oxidation process, as well as increasing electron emission rate, etc.[34,35]. However, to the best of our knowledge, the effects of multispecies coadsorption have not been investigated or applied in graphene CVD growth. In addition, during graphene CVD growth in a dichloromethane system, the high-electronegativity Cl can facilitate H detachment at graphene edges to enlarge domains. In this way, the use of dichloromethane can simultaneously accelerate the formation of active carbon species, nucleation, and coalescence of graphene, as well as the formation of continuous graphene films. Consequently, the growth rate of graphene on GFF was increased by ~3 orders of magnitude accompanied by ~960 times increase of carbon utilization, compared with the conventional methane-precursor system, and the full-coverage graphene film on GFF could be accomplished within ~0.5 min. The commercially available raw materials, i.e., dichloromethane and glass fiber, as well as the facile and efficient preparation strategy of GGFF, provided reliable premises for the cost-effective and energy-saving mass production of this new material and the solid foundation for the subsequent practical applications. Notably, GGFF is featured with excellent electrical conduction because of graphene covering, and the unique woven structure composed of warp and weft yarns (each containing thousands of fibers). The advantaged hierarchical conductive configuration of GGFF enables the ultrasensitive resistance response under pressure. GGFF thus showed promising performances as the lightweight, flexible pressure sensors with high sensitivity and portability used in human motion and physiological signal monitoring, such as pulse and vocal signals.

## Results

### Preparation of GGFF

The preparation process of GGFF through the direct CVD graphene growth on GGF is schematically presented in Fig. 1a, where dichloromethane was creatively introduced into the CVD system as the carbon precursor (see more details in Supplementary Fig. 1a and "Methods" section of Dichloromethane precursor-based CVD growth of GGFF). At the initial stage, dichloromethane undergoes the decomposition process to form the active carbon species (CH$_2$Cl, CH$_2$, CH, C, etc.) and Cl in the vapor phase. The active carbon species are subsequently absorbed onto the surface of GFF, leading to graphene nucleation. With the extension of growth time, active carbon species keep on attaching at the edges of graphene domains, resulting in their growth and coalescence to form continuous graphene films. Notably, during the above graphene CVD growth, Cl radicals produced by the decomposition of dichloromethane played important roles in facilitating the adsorption of active carbon species and the further graphene edge growth, which will be further discussed in Fig. 3.

The commercial GFF (with fiber diameters of ~7 μm) used for the preparation of GGFF has good high-temperature resistance, and the fiber shape was well maintained after ~1100 °C graphene CVD growth (Supplementary Fig. 2). Figure 1b presents the photographs of large-area GFF and GGFFs (with a width of ~25 cm) obtained with different growth times. It can be observed that contrasts of GGFFs gradually became darker with growth time extending, indicating the increase of graphene layer thickness, which is further supported by the decreasing intensity ratios of 2D and G peaks ($I_{2D}/I_G$) in Raman characterizations in Fig. 1c[36]. Notably, GGFF grown for ~0.5 min shows the typical Raman signal of monolayer graphene on insulating substrates ($I_{2D}/I_G$ = 1.5)[25,37], indicating the quite fast growth rate of graphene in this growth system. As revealed in Supplementary Fig. 3, I2D mappings of the fabricated GGFF with varying graphene thicknesses provided visual representations of high-layer uniformity. Cross-sectional transmission electron microscopy (TEM) images of GGFF further verified the nice layered structure of as-grown graphene (Supplementary Fig. 4). The graphene ribbon was obtained by etching the GFF core in hydrofluoric acid and graphene layers collapsing onto the silicon substrate, as displayed in scanning electron microscopy (SEM) image in Fig. 1d, and the corresponding energy-dispersive X-ray mapping of the ribbon was shown in Supplementary Fig. 5. In this way, the thickness of graphene film can be directly measured by atomic force microscopy (AFM). As presented in Fig. 1e, ~2 nm thickness of the graphene ribbon (measured after being transferred onto the silicon substrate), twice the thickness of the grown graphene layer, corresponds to 1–2 layers of the CVD-grown graphene[38]. Moreover, the AFM characterizations of graphene ribbons with different thicknesses obtained through growth time modulation were also provided in Supplementary Fig. 6, suggesting a good capacity for layer thickness control in our dichloromethane CVD growth system. GGFF inherits the excellent electrical conductivity of graphene. The sheet resistance mapping on GGFF in Fig. 1f reveals the high uniformity of the electrical conductivity with a mean value of $35.0 \pm 2.3\ \Omega\ sq^{-1}$ over the 5 cm × 5 cm area (see more details in the

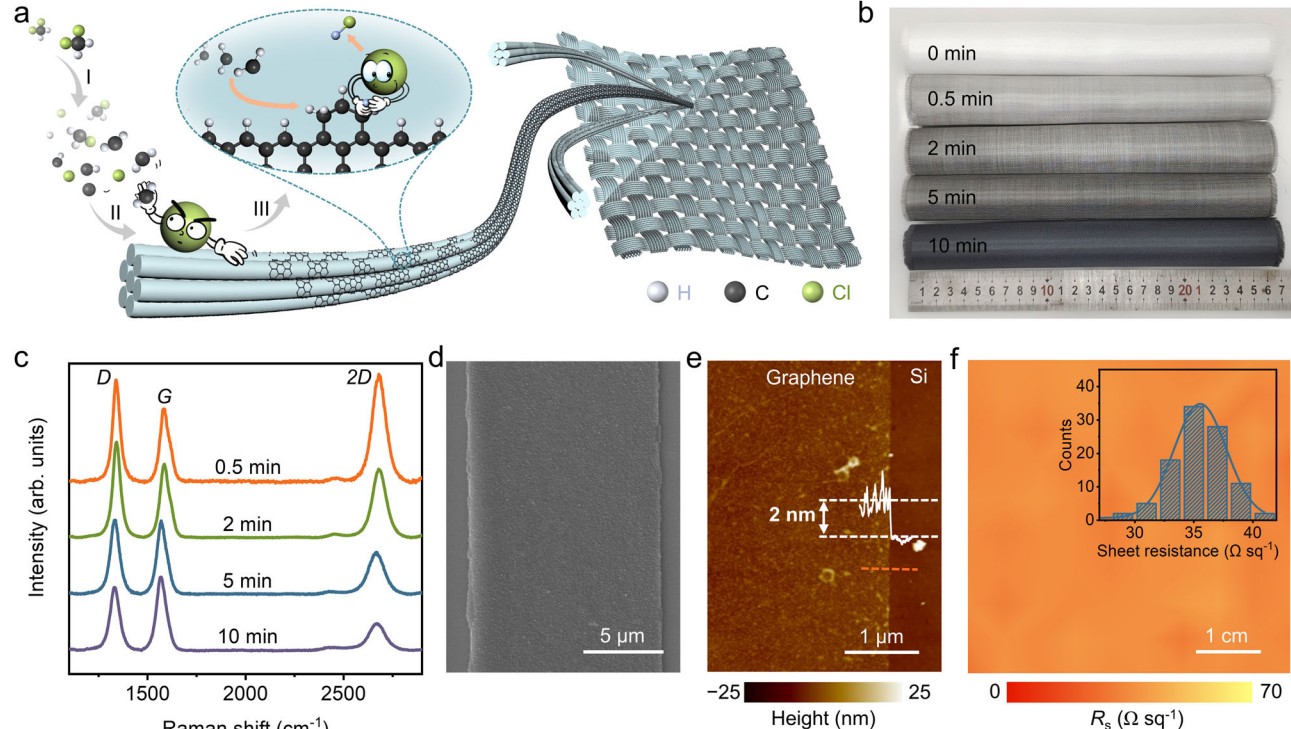

**Fig. 1 | Preparation of GGFF. a** Schematic of the graphene CVD growing process on GFF with dichloromethane as the precursor. Step I: thermal decomposition of dichloromethane in the vapor phase, step II: adsorption and nucleation of active carbon species on GGF, and step III: growth and coalescence of graphene domains by carbon attachment. **b** Photographs of large-area bare GFF and GGFFs (with a width of ~25 cm) obtained with the graphene growth times of 0.5, 2, 5, and 10 min (growth parameters: $H_2/CH_2Cl_2$ ratio of 8:1). **c** Raman spectra corresponding to GGFFs in (**b**) (normalized to G peak intensity). **d** SEM image of the graphene ribbon transferred onto the silicon substrate. The tube-shaped graphene on GGF collapsed into a ribbon after dissolving the GFF core. **e** Atomic force microscopy (AFM) image and height profile along the dashed orange line of the graphene ribbon transferred onto the silicon substrate, showing ribbon thickness of ~2 nm (twice the thickness of the grown graphene films), corresponding to 1–2 layers of graphene. **f** Sheet resistance mapping of GGFF (5 cm × 5 cm). Inset: corresponding statistics of sheet resistance.

"Methods" section of dichloromethane precursor-based CVD growth of GGFF).

### High graphene growth rate brought by dichloromethane carbon precursor

As schematically demonstrated in Fig. 2a, a four-point probe system was used for measuring the sheet resistance of GGFF. At the period for the formation of isolated graphene domains (period I in Fig. 2a), the sheet resistance was out of detection range due to the open-circuit condition in GGFF; after the continuous graphene films being formed, the conductive circuit was connected in GGFF and the sheet resistance could be successfully detected (period II in Fig. 2a) (see Supplementary Fig. 7 for current–voltage curves). Therefore, the detectability of the sheet resistance in GGFF could be regarded as the indicator for the formation of continuous graphene films.

The sheet resistance of GGFF as a function of growth time obtained with dichloromethane and methane precursors was systematically compared in Fig. 2b (carbon supplies introduced into the two CVD systems remained consistent) (see more details in the "Methods" section of Dichloromethane precursor-based CVD growth of GGFF and Methane precursor-based CVD growth of GGFF under consistent carbon supplies with dichloromethane-based growth). It can be observed that the sheet resistances decreased with growth time extending for both GGFFs, implying the improvement of electrical conduction. This will satisfy the various application scenarios, which have different requirements for electrical conductivity. Notably, in the dichloromethane system, the full-coverage continuous graphene films could be formed within ~0.5 min, in contrast to ~480 min taken in the methane system (the pentagram points marked in Fig. 2b), and the

carbon utilization was therefore largely improved by ~960 times. Moreover, to obtain the same electrical conduction of GGFF, the dichloromethane growing system required significantly less time compared with methane. For instance, to reach the target sheet resistance of ~30 Ω sq$^{-1}$ (the triangle points marked in Fig. 2b), it only required ~10 min for dichloromethane, while ~900 min was needed for methane. The growth rate ($v$, min$^{-1}$) can be determined as graphene coverage ($\theta$) divided by growth time ($t$): $v = \theta/t$. When the full coverage is achieved, the growth rate can be expressed as $v = 1/t$. To the best of our knowledge, the graphene growth rate obtained in this work is much higher than that in reported literature for graphene grown on nonmetallic substrates, even on metal substrates such as Ni and Ni−Cu alloy[21,25,39–44], as summarized in Fig. 2c. For different growing conditions, such as hydrogen-to-dichloromethane ($H_2/CH_2Cl_2$) or hydrogen-to-methane ($H_2/CH_4$) ratios of 5, 8, 16, 20, and 25, the growth rates of graphene on GFF in the dichloromethane system also kept at ~3 orders of magnitude higher that of methane (Supplementary Fig. 8), verifying the effectiveness of dichloromethane precursor for accelerating graphene growth. These results reveal the prominent advantages of dichloromethane precursor in production capacity improvement and energy saving (shorting high-temperature treatment time) during the graphene CVD mass production.

In addition, a dichloromethane CVD growth system can achieve high crystal quality under the premise of a high growth rate. There is usually a dilemma between the growth rate and quality. Figure 2d, e presents the TEM images on TEM grid of dichloromethane- and methane-grown graphene obtained at the same growth rate (~2 min$^{-1}$) (see more details in Supplementary Fig. 9 and "Methods" section of Dichloromethane precursor-based CVD growth of GGFF and Methane

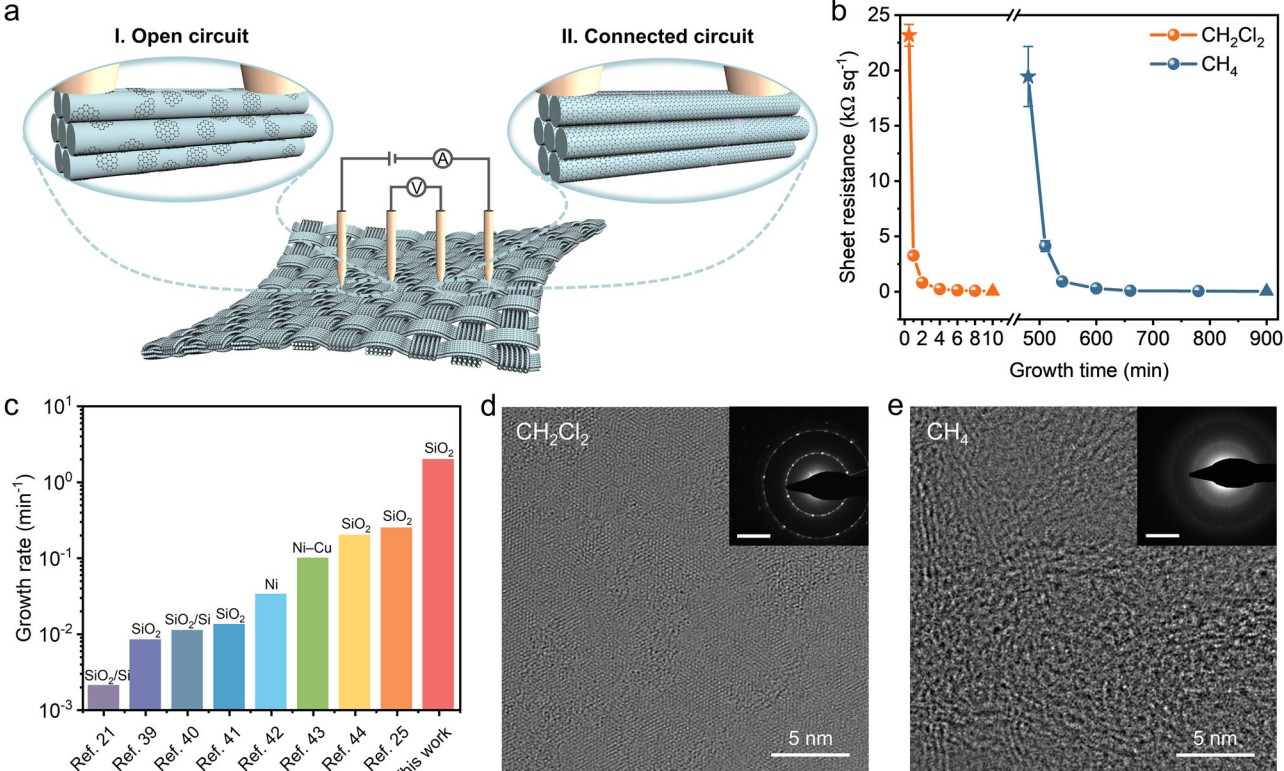

**Fig. 2 | Rapid growth of graphene on GFF with dichloromethane as the carbon precursor. a** Schematic depicting the sheet resistance measurements of GGFF at different graphene growing periods. Period I: open circuit condition between isolated graphene domains, period II: connected circuit condition of continuous graphene films. **b** Sheet resistance as a function of growth time for GFFFs prepared using dichloromethane (orange) and methane (blue) precursors. The carbon supplies remained consistent in both systems. The pentagram points marked the time for the formation of full-coverage graphene films and the triangle points marked the time for reaching the sheet resistance of ~30 Ω sq$^{-1}$. The error bars represent the standard deviations ($n = 5$). **c** Comparisons between the growth rates of graphene obtained in this work with those reported in literature[21,25,39–44]. **d, e** High-resolution TEM images on a TEM grid of dichloromethane- and methane-grown graphene obtained under the same graphene growth rate (growth parameters: $H_2/CH_2Cl_2$ ratio of 8:1; $H_2/CH_4$ ratio of 1:5). Insets are corresponding selected area electron diffraction (SAED) patterns (scale bars: 5 n m$^{-1}$).

precursor-based CVD growth of GGFF under consistent growth rate with dichloromethane-based growth), which reveal the longer-range crystalline order of dichloromethane-grown graphene in contrast to the amorphous structure of the methane-grown graphene. Consistently, in contrast to the obvious characteristic Raman 2D band (~2680 cm$^{-1}$) of dichloromethane-grown graphene, the 2D band for methane-grown graphene is nearly negligible (Supplementary Fig. 10), indicating the extremely low crystal quality of the formed graphene films[45]. These results suggest that dichloromethane is indeed a promising carbon precursor for rapid graphene growth with satisfactory crystallinity.

**Mechanisms for the rapid growth behaviors in dichloromethane CVD system**

To reveal the underlying mechanisms of the greatly increased growth rate of graphene on GFF in a dichloromethane CVD system, the density function theory (DFT) calculations about the molecular characteristics of dichloromethane and its participation in the elementary processes of graphene CVD growth were carried out. As shown in Supplementary Fig. 11, the C−Cl bond strength in dichloromethane is much weaker than the C−H bond in methane due to the longer bond length (~1.75 Å for C−Cl vs ~1.13 Å for C−H) and smaller overlap populations (~0.42 for C−Cl vs ~0.77 for C−H). Therefore, the dichloromethane precursor appears to be more prone to pyrolysis in the gas phase. This was further verified by the kinetic calculations of dichloromethane pyrolysis in Fig. 3a and Supplementary Fig. 12. Due to the much lower

energy barrier of dichloromethane dechlorinating than that of methane dehydrogenating, $CH_2$ is the dominant active species in dichloromethane system and is more abundant in concentration under the experimental temperature of ~1100 °C (Supplementary Fig. 13), instead of $CH_3$ as that in methane system[46]. Benefiting from the higher reaction activity than $CH_3$, the dominant $CH_2$ species will largely facilitate the subsequent growth of graphene on GFF.

Further, the interaction between the pyrolysis products of dichloromethane and GFF substrate (~99.9% $SiO_2$) was explored. A dominant (0001) orientation of the $SiO_2$ surface was chosen as the typical calculation model of the GFF substrate based on its relatively higher thermodynamic stability[47]. Considering the H-rich graphene CVD growing conditions, O1-, O2-, and Si-terminated surface structures with or without H passivation were all considered during the calculations, as schematically shown in Supplementary Fig. 14. The surface energy calculations of the six above configurations revealed that Si-terminated $SiO_2$(0001) without H passivation was the most stable structure with the lowest surface energy (Supplementary Fig. 14). Therefore, GFF with the Si-terminated $SiO_2$(0001) without H passivation was selected as the calculated model, and in this model there are two types of sites (Si site and O site) for $CH_2$ species adsorption. The adsorption energy of $CH_2$ is ~−1.24 eV on the Si site and −1.40 eV on the O site, which reveals the nonactivity of O site with the unstable $CH_2$ adsorption (Supplementary Fig. 15 and Supplementary Table 1). Notably, when $CH_2$ and Cl are coadsorbed, the redistribution of the electronic density at the surface occurs. As shown in Fig. 3b, the

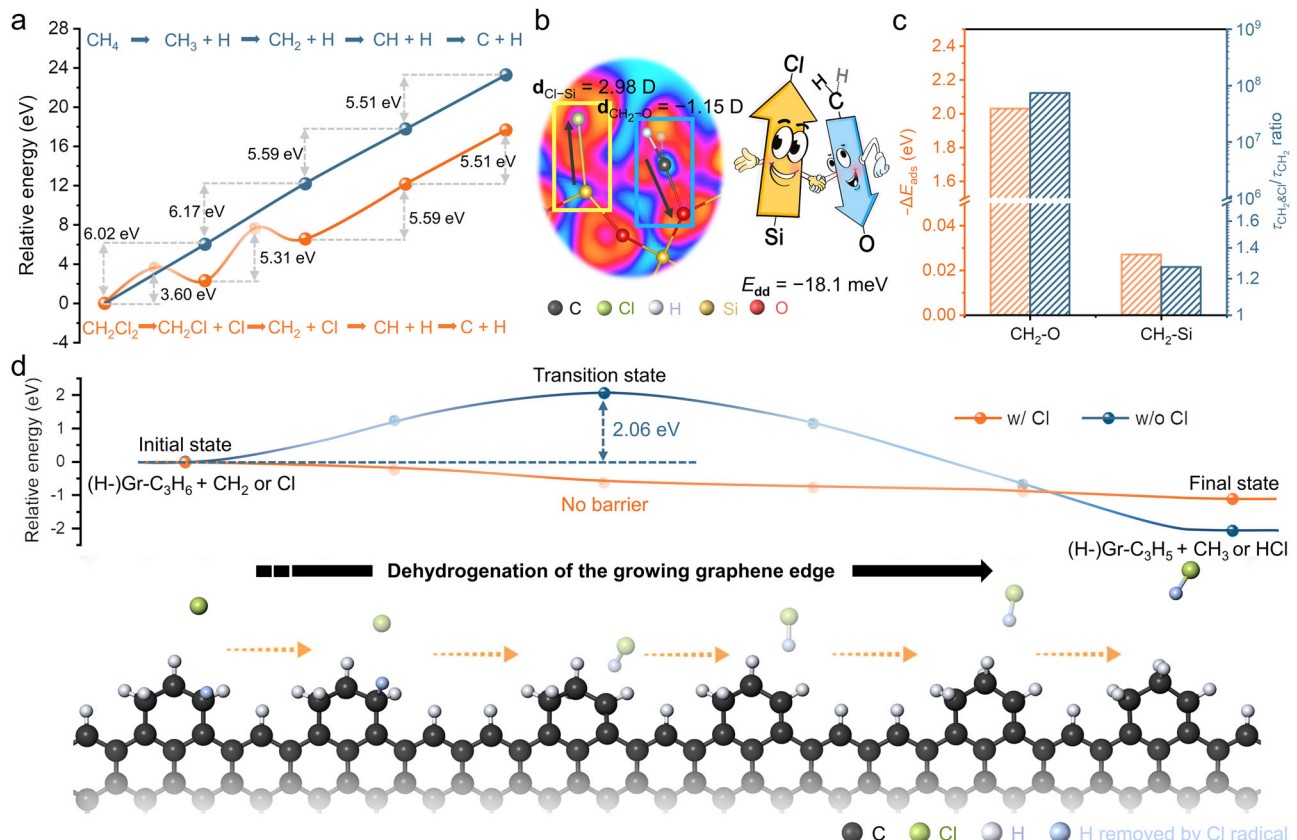

**Fig. 3 | Theoretical investigations using density function theory (DFT) of the rapid growth mechanisms of graphene on glass fiber fabric (GFF) in dichloromethane chemical vapor deposition (CVD) system. a** Energy profiles during the full pyrolysis of dichloromethane and methane in the gas phase. **b** Dipole-dipole interaction resulting from the multispecies-coadsorption of Cl on Si site and $CH_2$ on the adjacent non-bonding O site. **c** Changes of adsorption energy ($E_{ads}$) and lifetime ($\tau$) of dominant $CH_2$ species at O or Si adsorption sites on Si-terminated $SiO_2(0001)$ without H passivation before and after Cl coadsorption ($\Delta E_{ads} = -(E_{ads}(CH_2$ and Cl) $- E_{ads}(CH_2))$). **d** Energy profiles for the dehydrogenation and reconfiguration kinetics of the growing graphene zigzag edges with (blue) or without (orange) Cl participation, where (H-) Gr represents that the frontier edge of graphene is terminated by H atoms. Bottom schematic showing graphene edge dehydrogenation processes with Cl assisting.

electronegative Cl absorbed on Si site draws away electrons from Si, leading to a positive surface dipole moment $\mathbf{d}_{Cl-Si} = 2.98$ D. In contrast, the electropositive $CH_2$ absorbed on the adjacent non-bonding O site contributes electrons to O, resulting in a negative surface dipole moment $\mathbf{d}_{CH_2-O} = -1.15$ D. Therefore, an attractive dipole–dipole interaction is caused between these two opposite surface dipoles with the interaction energy $E_{dd} = \mathbf{d}_{Cl-Si} \ \mathbf{d}_{CH_2-O}/4\pi\varepsilon_0 r^{-3} = -18.1$ meV[35,48,49], which reduces $CH_2$ adsorption energy by -2.0 eV and increases the adsorption life by ~8 orders of magnitude (Fig. 3c). Therefore, the inactive O sites on $SiO_2(0001)$ surface are greatly activated by the coadsorbed Cl. In addition, the activity of Si sites is also slightly enhanced with a decreasing $CH_2$ adsorption energy and an increasing adsorption lifetime (Fig. 3c). Consequently, due to the Cl–$CH_2$ coadsorption, the capacity for GFF substrate to capture active carbon species is largely enhanced, which is of great significance to promote the nucleation and growth processes of graphene on the substrate.

Highly electronegative Cl radical, as one of the significant decomposed products of dichloromethane in graphene CVD system, played an important role in the growth of graphene domains[50]. After the $CH_2$ species attached at the growing edges of graphene domains (Supplementary Fig. 16), their dehydrogenation subsequently happened to change the configuration from $sp^3$ hybridization to $sp^2$ hybridization, and to prepare for the bonding of the next $CH_2$ species. Conventionally, the H atoms at graphene edges are removed by hydrocarbon species in the gas phase via $CH_x + H - Gr \rightarrow CH_{x+1} + Gr$ reaction, but the high reaction energy barrier severely limits the

expansion rate of graphene edges (-2.06 eV for $CH_2$ (blue line in Fig. 3d, Supplementary Fig. 17 and Supplementary Table 2), -1.94 eV for $CH_3$ as previously reported[51]). In contrast, since the produced Cl radicals in the dichloromethane system possess high electronegativity, they can spontaneously react with the H atoms bonded at the growing edges of graphene domains (energy barrier-free, orange line in Fig. 3d). Therefore, the edges expansion rate of graphene domains on GGF is greatly increased, which will largely accelerate the coalescence of graphene domains and the formation of continuous graphene films.

The difference in the rate ($\nu$) of graphene growth using $CH_4$ and $CH_2Cl_2$ can be comprehensively evaluated by the following equation:

$$\nu = \frac{\Delta L \times C_p}{e^{(E_a(RL) + \Delta E)/k_B T}} \qquad (1)$$

where the $E_a$ (RL) and $\Delta E$ take the values of the threshold barriers during graphene growth and the reaction heat for dehydrogenation of the graphene growing edge, respectively. $\Delta L = 0.142$ nm is the length of a C–C bond in graphene, and $C_P$ represents the collision rate of active carbon species[51], which is proportional to the partial pressure of the carbon species in the gas phase (see more details in Supplementary Fig. 13). Therefore, after substituting the values into above equation, it can be obtained that the rate of $CH_2Cl_2$ involved in the growth of graphene is about $10^3$ times higher than that of $CH_4$, which is essentially compatible with the experimental finding shown in Supplementary Fig. 8.

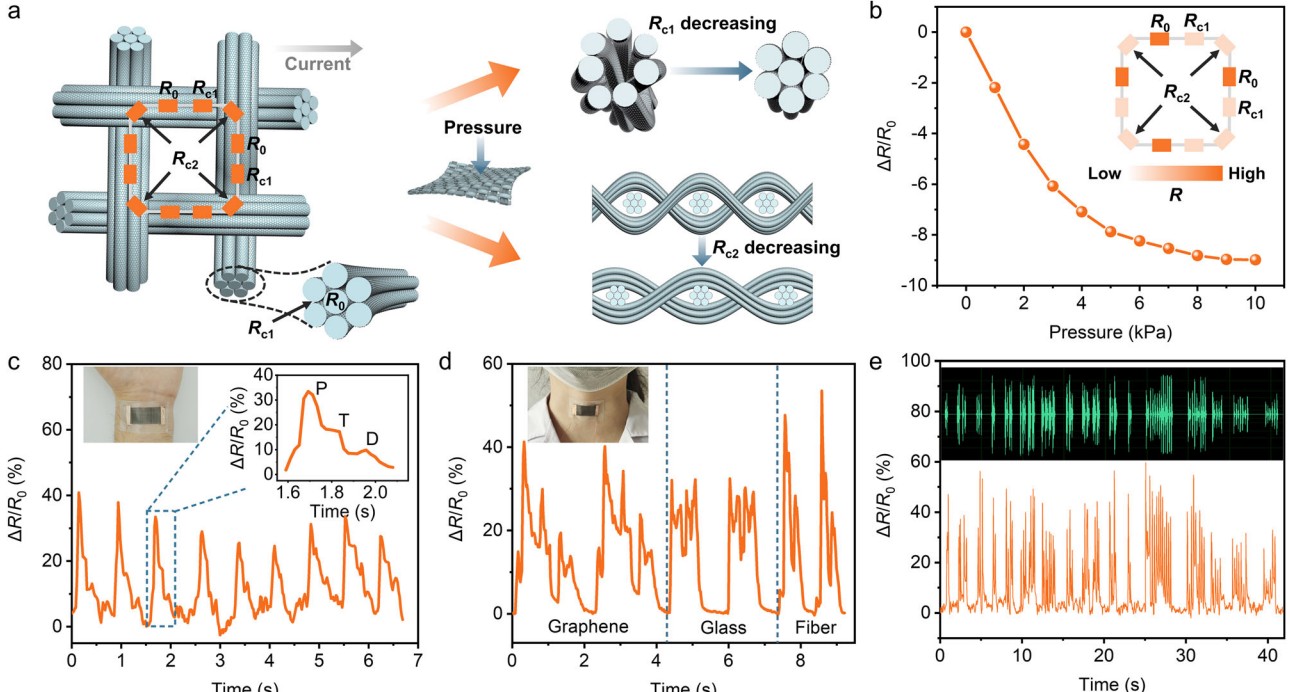

**Fig. 4 | GGFF flexible pressure sensors. a** Schematic for the elementary unit of GGFF and corresponding equivalent circuit (inset), depicting the hierarchical conductive model constructed in GGFF. $R_0$ is the intrinsic resistance of each fiber, $R_{c1}$ is the contact resistance between neighboring fibers, and $R_{c2}$ is the contact resistance between the warp and weft yarns. **b** Relative resistance variation under different pressure contact resistance between the warp and weft yarns. Inset: corresponding equivalent circuit under the pressure, where the faded orange color of $R_{c1}$ and $R_{c2}$ denotes the resistance decreasing. **c** Pulse signal detection with GGFF sensor. Inset: wrist attached with GGFF sensor (left), and the magnified image of a single pulse presenting the typical P-, T-, and D-waves (right). **d** Human vocal signal detection with GGFF sensor, presenting the featured peak shapes responding to the phonation of "graphene", "glass", and "fiber" with high repeatability. Inset: throat attached with GGFF sensor. **e** $\Delta R/R_0$ of GGFF sensor attached on a loudspeaker playing an audio file (bottom), showing good consistency with the original audio signals (up).

## GGFF flexible pressure sensor based on the hierarchical conductive configuration

After graphene growth on GFF substrates, a 3D conductive network came into being where graphene glass fiber in the fabric served as the conductive channel. Figure 4a schematically illustrates the elementary unit of GGFF and the corresponding equivalent circuit (inset in Fig. 4a). Since GGFF is composed of warp and weft yarns and each yarn contains thousands of fibers, the resistance of GGFF is determined by the intrinsic resistance of each fiber ($R_0$), contact resistance between neighboring fibers ($R_{c1}$), and contact resistance between the warp and weft yarns ($R_{c2}$). Therefore, a hierarchical conductive pathway in GGFF is constructed with the basic resistance parameters of $R_0$, $R_{c1}$, and $R_{c2}$. Among them, the contact resistance ($R_{c1}$ and $R_{c2}$) can be evaluated via the Holm's theory[52], using

$$R_c = \frac{\rho}{2}\sqrt{\frac{\pi H}{nP}} \tag{2}$$

where $\rho$ is the electrical resistivity, $H$ is the material hardness, $n$ is the number of contact points, and $P$ is the contact pressure. The mechanical deformation of GGFF under the pressure can change the number of contact points and contact pressure, leading to the variation of contact resistance $R_{c1}$ and $R_{c2}$ (Fig. 4a). As verified in Fig. 4b, the total circuit resistance of GGFF decreases when pressure was applied, related to the decreased $R_{c1}$ and $R_{c2}$ in the equivalent circuit because of the closer contacts between the neighboring fibers, as well as the warp and weft yarns (inset in Fig. 4b).

The hierarchical conductive configuration of GGFF enables the ultrasensitive response under external pressure, providing a promising platform for pressure sensor applications. As shown in Supplementary

Fig. 18, a flexible sensor was fabricated based on GGFF encapsulated with polypropylene films (see more details in the "Methods" section of fabrication of GGFF flexible pressure sensors). The resistance response to the finger bending was monitored, exhibiting rapid relative resistance variations ($\Delta R/R_0$) with good repeatability and stability (Supplementary Fig. 19). Besides the above large-range movement, the GGFF flexible sensor could capture the weak physiological signals. As shown in Fig. 4c, the sensor was attached to the wrist for detecting real-time pulse signals. Repeatable and regular pulse shapes were detected under the relaxation conditions, and the $\Delta R/R_0$ could reach ~40%, which is higher than most of the state-of-the-art sensors[53–56]. Furthermore, each pulse peak clearly displays the typical features of the pulse waveform, i.e., percussion wave (P wave), tidal wave (T wave), and diastolic wave (D wave) (inset in Fig. 4c)[57], indicating the high sensitivity of the sensor. Human vocal signals could also be recognized with the GGFF sensor responding to the throat muscle motion. As presented in Fig. 4d, the words "graphene", "glass", and "fiber" can be recorded in different patterns with excellent repeatability, which endows the GGFF sensor with the capacity for human sound collection and recognition[58]. In addition, Fig. 4e presented the collected $\Delta R/R_0$ when the GGFF sensor was attached to a loudspeaker playing a burst of birdsong (Supplementary Movie 1). It was found that the detected signals had a synchronous response to the original audio frequency, and could retain almost every characteristic peak. In this way, GGFF can be highly expected in sound visualization technology, such as mobile health care, fatigue detection, and robotic voice development[59,60]. Applied as flexible sensors, the performance stability under various mechanical deformations was the significant premise. As shown in Supplementary Fig. 20, after repeated deformations of twisting, grasping, and folding, the morphology of GFFF presented

negligible change and no peeling of graphene layers was observed, suggesting the high flexibility and interfacial stability of GGFF. Overall, the sensitive motion-resistance response, the excellent flexibility, and the light weight make GGFF a promising candidate as the highly portable pressure sensor for human motion and physiological signal monitoring.

Notably, GGFF's large-area and scalable production capabilities position it uniquely for large-size applications in various areas. For example, as we know, the aircraft wing sensors are the essential parts for ensuring the normal work of aircraft. These sensors are anticipated to play a pivotal role in monitoring various parameters critical to an aircraft's wing, such as the strain and deformation, temperature, as well as structural integrity[31,61,62]. GGFF holds promising application potential in the above scenarios, i.e., aircraft wing sensors. First, according to the analyses in Fig. 4, the GGFF-based sensor exhibited high sensitivity for the resistance response. Second, GGFF presented excellent flexibility, which can realize a conformal fit with objects of different shapes to realize effective signal acquisitions. Therefore, in the fields of aircraft sensors, the large-area GGFF will present the expected application values. Beyond sensors, GGFF also exhibited exceptional electrical heating performances, which made it a promising electric heating material used in the areas of anti/de-icing of large instruments or equipment, such as aircraft and wind turbine blades. In addition, the excellent structural flexibility of GGFF allows it to conform seamlessly to various surfaces, ensuring comprehensive anti/de-icing protection across expansive areas. Moreover, GGFF has a low density of ~2.5 g cm⁻³, which avoids the additional weight gain for the aircraft and wind turbine blades during large-area practical applications. The large-area coverage, high production capacity, excellent flexibility, and lightweight, as well as excellent tolerance to harsh environments, make GGFF a superior anti/de-icing material for aircraft and wind turbine blades, which will inject new impetus into the development of aviation and renewable energy areas.

## Discussion

In this work, GGFF with the innovative hierarchical conductive configuration was successfully developed, where the multi-elementary-process modulation, especially the multispecies-coadsorption, was conducted to realize the rapid growth of graphene on the insulating GFF substrate. Dichloromethane, a widely used organic solvent in industry, applied as the carbon precursor for graphene CVD growth possessed a low decomposition energy barrier so as to produce rich active carbon species, and more importantly, the produced highly electronegative Cl in this CVD system enhanced the adsorption of active carbon species by Cl–CH₂ co-adsorption and facilitate H detachment from graphene edges, which largely promoted the adsorption and nucleation of graphene on the substrate and their further growing coalescence to form continuous graphene films, respectively. Notably, the Cl–CH₂ co-adsorption strategy was first proposed in the graphene CVD research area. We noted that tri-chloromethane and dichloromethane were previously introduced into a plasma-enhanced CVD (PECVD) system to promote the decomposition of mixed precursors[50]. However, the implementation of dichloromethane in conventional thermal CVD systems represents a significant advancement since the intrinsic decomposition properties of precursors play a much more vital role in the thermal CVD process than in the PECVD process. The commercially available raw materials, i.e., dichloromethane and glass fiber, as well as the facile and efficient preparation strategy, provided reliable premises for the cost-effective and energy-saving mass production of GGFF. GGFF is featured with the hierarchical conductive configuration constructed with warp and weft yarns consisting of thousands of fibers, which enables the ultra-sensitive resistance response under pressure. In this way, GGFF shows promising potential as lightweight, flexible sensors with high

sensitivity and portability used in human motion and physiological signal monitoring. As is known to all, the trade-off between high quality and high growth rate for graphene CVD growth on noncatalytic non-metallic substrates is a recognized issue for CVD graphene, which is also the direction we are committed to in future research. Beyond quality considerations, the industrial applicability of graphene hinges on factors such as production capacity and cost. Our approach offers an efficient and cost-effective solution to mitigate the high energy consumption associated with the prolonged high-temperature CVD growth process of graphene in the noncatalytic system, thereby addressing critical concerns in the mass production of CVD graphene.

## Methods
### Ethics declarations
The data were obtained with the informed consent of all participants. All human experiments were performed in compliance with the protocol approved by the Institutional Review Board of Tsinghua University (no. 20230019).

### Dichloromethane precursor-based CVD growth of GGFF
Commercially available GFF of ~0.1 mm thickness (Wuhan Sino Type Optoelectronic Technology CO., LTD) was first annealed under ~500 °C in ambient air for 2 h to remove the coated polymer (same treatment for below). After being carefully cleaned, GFF was placed at the center of a high-temperature furnace with the low-pressure CVD (LPCVD) system. In a typical procedure, the system was evacuated to a base pressure of <1 Pa and heated to 1100 °C under a H₂ flow of 120 sccm. Subsequently, dichloromethane vapor with the desired flow (5–24 sccm) was pumped into the chamber for graphene growth. Throughout the growth process, the chamber pressure was held at approximately 400–500 Pa depending on the flow of dichloromethane vapor. The growth of graphene lasted for 0.5–10 min, followed by the natural cooling process to room temperature under an H₂ flow of 20 sccm and Ar flow of 200 sccm. The temperature profile diagrams of the CVD process and information about the flow rate of precursors are shown in Supplementary Fig. 1a[63,64]. GGFF samples for sheet resistance mapping (Fig. 1f) and TEM characterization (Fig. 2d) were grown for ~10 min and 0.5 min, respectively, with the H₂ flow of 120 sccm and the dichloromethane vapor flow of 15 sccm at ~1100 °C.

### Methane precursor-based CVD growth of GGFF under consistent carbon supplies with dichloromethane-based growth
The same GFF substrates and LPCVD method were used in experiments. The system was evacuated to a base pressure of <1 Pa and heated to ~1100 °C under a H₂ flow of 120 sccm. Subsequently, methane gas with the desired flow (5–24 sccm) was introduced to the chamber for graphene growth. Throughout the growth process, the chamber pressure was held at approximately 400–500 Pa depending on the flow of methane. In each individual comparative experiment, the methane flow remained consistent with the flow of dichloromethane vapor to ensure consistency across the comparative experiments. The growth of graphene lasted for 8–15 h, followed by the natural cooling process to room temperature under an H₂ flow of 20 sccm and Ar flow of 200 sccm. The temperature profile diagrams of the CVD process and information about the flow rate of precursors are shown in Supplementary Fig. 1b[63,64].

### Methane precursor-based CVD growth of GGFF under consistent growth rate with dichloromethane-based growth
To obtain the same growth rate of methane-grown graphene (Fig. 2e) as that of dichloromethane-grown graphene, the H₂/CH₄ ratio was further reduced to 1:5 with the H₂ flow of 10 sccm and the methane flow of 50 sccm. The growth temperature was maintained at ~1100 °C and the CVD growth process lasted for ~0.5 min.

## Fabrication of GGFF flexible pressure sensors

GGFF for the pressure sensor fabrication was synthesized with the $H_2$ flow of 120 sccm and the methane flow of 15 sccm at 1100 °C for ~4 min. Subsequently, copper double-sided tapes were stuck on both ends of GGFF to serve as the electrodes. Also, copper wires were connected to make an electrical connection. Finally, polypropylene films tightened by a hot plate were used to package the sensor. Two volunteers participated in the GGFF pressure sensor experiments, comprising a 27-year-old male for finger bending and pulse signal detections and a 26-year-old female for human vocal signal detection. The gender of participants was determined based on self-report. Gender was not considered in the study design because the sensitivity of pressure sensors has no certain relationship with gender.

## Graphene transfer for AFM and TEM characterizations

The graphene transfer process for AFM and TEM characterizations was carried out with the assistance of polydimethylsiloxane (PDMS) stamps[65]. First, GGFF was put on a PDMS stamp and flattened with a glass plate. After the adhesion of PDMS, the GGFF/PDMS assembly was immersed in hydrofluoric acid solution (20 wt%) for 8 h to etch the GFF substrate, followed by a repeated rinsing process with deionized water. The resulting graphene/PDMS assembly was then pressed onto the target substrate (silicon for AFM and TEM grid for TEM) at 80 °C for 2 h. Finally, the PDMS stamp was carefully peeled off from the substrate, resulting in the transferred graphene layer on the target substrate.

## Characterization

The prepared samples were characterized using SEM (Thermo Scientific Quattro S, acceleration voltage: 10 kV), Raman spectroscopy (Horiba, LabRAM HR800, 532 nm laser wavelength), AFM (Bruker Dimension Icon with ScanAsyst mode), and TEM (FEI Tecnai F20, acceleration voltage: 200 kV). The sheet resistance was measured by a four-point probe resistivity measurement system (RTS-8). The electrical signals of sensors were recorded with a Keithley 2450 digital Sourcemeter at a constant voltage of 0.1 V.

## DFT computational details

The DFT calculations are performed using the Vienna Ab initio Simulation Package software[66]. The projector-augmented wave method was used to describe the interactions between the core and valence electrons. The Perdew–Burke–Ernzerhof form involving the generalized gradient approximation functional was used to describe the exchange-correlation potential[67]. The electron wave functions were expanded in a plane wave basis set with a kinetic energy cutoff of 450 eV, the Monkhorst–Pack scheme $k$-point grids were set to $2 \times 2 \times 1$. A vacuum layer of 15 Å is added perpendicular to the sheet to avoid artificial interaction between periodic images. All the structures are relaxed until the residual forces on the atoms have declined to less than $0.05$ eV Å$^{-1}$, and the convergence criterion for the total energy is set to $1 \times 10^{-4}$ eV. The calculation of the atomic dipole moments, with consideration of periodic boundary conditions, was accomplished by the application of the recently developed density-derived electrostatic and chemical methods[68]. For transition state calculations, the climbing image nudged elastic band method was used to determine the energy barriers of various kinetic processes[69]. The transition states during each elementary reaction were verified by transition state imaginary frequency calculations.

The stability of hydrocarbon species adsorption geometry was evaluated by the adsorption energy ($E_{ads}$), which is defined as

$$E_{ads} = E_{tot} - E_{sub} - E_{CH_x} \tag{3}$$

where $E_{tot}$ and $E_{sub}$ are the total energies of the $SiO_2$ (0001) substrate with and without adsorbed hydrocarbon species, $E_{CH_x}$ is the energy of a hydrocarbon molecule.

During the various elementary reaction calculations and transition state searching processes, the reaction barrier $E_a$ is calculated as

$$E_a = E_{TS} - E_{IS} \tag{4}$$

where $E_{IS}$ and $E_{TS}$ represent the energies of the initial state (IS) and transition state (TS), respectively.

Considering the weak absorption on insulating surfaces, the lifetime of a species on a surface can be estimated by

$$\tau = \tau_0 \exp(E_b/k_B T) \tag{5}$$

where $\tau_0 = h/k_B T$ is the prefactor and $h$ and $k_B$ are the Plank and Boltzmann constants, respectively.

## Reporting summary

Further information on research design is available in the Nature Portfolio Reporting Summary linked to this article.

## Data availability

Source data are provided with this paper. All other data that support the plots within this paper and other findings of this study are available from the Supplementary Information or the corresponding authors upon request. Source data are provided with this paper.

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

## Acknowledgements

This work was financially supported by the National Natural Science Foundation of China (NSFC, nos. 52272032, T2188101, and 52021006 to Z.F.L.), and the Beijing Nova Program of Science and Technology (no. 20220484079 to Y.Q.).

## Author contributions

Z.L., Y.Q., and K.W. conceived and designed the experiments. Z.L. and Y.Q. supervised the project. K.W. designed and developed the dichloromethane precursor-based CVD system for the rapid preparation of GGFF. S.C., Y.C., K.H., R.L., H.Y., W.L., F.L., Y.Y., F.Y., K.Z., Z.L., C.T., M.M., and Y.G. performed the SEM, AFM, TEM, Raman, and sheet resistance characterizations. K.W., X.S., and W.Y. studied the mechanism of rapid preparation of GGFF induced by multispecies coadsorption. K.W. and S.C. made the GGFF-based flexible pressure sensor and tested its sensitivity under M.L. and M.J.'s technical assistance. K.W., X.S., and S.C. performed the data analysis and wrote the manuscript under the guidance of Y.Q. and Z.L. All authors contributed to the discussion and analysis of the results.

## Competing interests

The authors declare no competing interests.
