## [Peer Review File · Nature Communications]

Multispecies-Coadsorption-Induced Rapid Preparation of Graphene Glass Fiber Fabric and Applications in Flexible Pressure SensorREVIEWER COMMENTS

Reviewer #1 (Remarks to the Author):

Wang et al. reported a method for directly growing graphene via CVD on glass fiber fabric. They employed the dichloromethane as a precursor to accelerate the graphene growth. The authors claimed that this approach increased the growth rate by approximately three orders of magnitude compared to the traditional methane precursor, significantly enhancing the graphene growth speed. In recent years, considerable research has been published on growing graphene on insulating substrates with different precursors. The use of dichloromethane in this work could be considered one of the optimizations for current growth techniques.

At its current state, frankly, I cannot recommend the publication due to numerous serious concerns. , I would be willing to reconsider my recommendation if the authors can address all my questions and make considerable improvements. Refer to the following suggestions.

Dichloromethane.

1. The authors claim, "the dichloromethane, a widely used organic solvent in industry, was first used as the carbon precursor for graphene CVD growth." It is not true. A quick online search would reveal related reports on dichloromethane's effect on graphene growth, such as one work by Prof. Ding et al. in *Advanced Science*, 2022, 9(15): 2200737. This claim requires reevaluation.
2. The authors conducted similar DFT calculations to the work mentioned above. Their calculation results also suggest that CHCl_3 and CCl_4 could be better choices than CH_2Cl_2 . The authors should comprehensively compare these choices and explain the advantages of using dichloromethane.

Experimental evidence.

1. The claim "the increase of graphene layer thickness, which are further supported by the increasing intensity ratios of 2D and G peaks (I_{2D}/I_G) in Raman characterizations in Fig. 1c." is incorrect. Refer to the work from Ruoff's group in *Nature Nanotechnology* volume 15, pages 289–295, 2020.
2. The lack of significant change in G-band intensity with substantial thickness increase is puzzling. Additionally, the 2D over G peak ratio appears not correct.
3. The excessive D-band intensity, even surpassing the G band, suggests poor graphene crystallinity. The decrease in D-band intensity as thickness increases needs detailed explanations.
4. Additional characterization.

Raman maps are required on various thicknesses of graphene to assess layer uniformity.

The authors should provide a detailed characterization of the graphene ribbon in Figure 1D. Better, including the EDX mapping in SEM characterization. The cross-sectional TEM of grown graphene is necessary to confirm their layered structure.

AFM image is confusing in Fig. 1b. Ensure accurate line profiles and provide AFM characterization for various thicknesses of graphene. Indicate the graphene region.

Comparison

1. The claim about the high energy consumption due to CVD growth temperature contradicts the use of 1100°C growth temperature in this work. Explain this.
2. The growth rate comparison seems inappropriate and meaningless, as it compares the rate of multilayer graphene with studies on monolayer graphene growth. In most cases of single-layer graphene growth, the primary goal is precisely controlling the layer thickness and uniformity. Refer to the work in *Advanced Materials*, 2019, 31(35): 1903615. Again, references 17, 18, 24, and 40 .. pertain to the growth of monolayer graphene. The authors are suggested to focus on comparing mass-produced multilayer graphene growth, whether on metals or glasses.

Growth mechanism.

1. The authors should provide detailed growth conditions, including temperature, chamber pressure, gas flows, and time, on various experiments for reproducibility.
2. The mechanism of graphene nucleation on fibers is required to be explained. Show individual graphene grains on fibers in SEM images.

Errors.

Numerous typos and errors in the text need correction. Wrong Figure number "Figure 3d, e."

This work is too premature for publication until all concerns are adequately addressed.

Reviewer #2 (Remarks to the Author):

The article submitted by Kun Wang et al. presents an intriguing study on the direct chemical vapor deposition (CVD) growth of graphene on dielectric/insulating materials, particularly focusing on glass fiber fabric as the substrate. The novel approach of using dichloromethane as the carbon precursor to promote and accelerate graphene growth on the noncatalytic nonmetallic substrates will be a promising route to break the bottleneck for graphene mass production and practical applications. The mechanisms behind this choice, particularly the low decomposition energy barrier of dichloromethane and its role in enhancing carbon species adsorption and hydrogen detachment at graphene edges, are explained clearly and logically. I believe this work will attract wide interests from the areas of graphene and 2D materials in the academia, as well as the attentions from the industry, who are concerned about the practical applications of graphene. Therefore, I recommend it can be accepted after a minor revision emphasizing the following questions.

1. The authors demonstrated the rapid growth of graphene in a dichloromethane system to achieve high electrical conductivity. It will be better to provide more detailed application scenarios where the conductive graphene glass fiber fabric is advantageous.
2. The author claimed the high flexibility of the fabricated graphene glass fiber fabric, but didn't present the actual photos of the fabric being grasped, folded, twisted etc., and the morphology of graphene on the fiber after the mechanical deformations, is there any detachment or peeling happened? Suggest to provide more information.
3. The AFM image in Fig. 1e revealed an interlayer distance of graphene in the collapsed ribbon that appears relatively larger than the intrinsic value of graphene (~ 0.34 nm). Can the authors provide insights into the reasons behind this observed difference in interlayer distance?
4. Given that chlorine is produced by dichloromethane in the CVD system, there's a concern about potential chlorine doping of the as-grown graphene. Suggest investigating whether the as-grown graphene is indeed doped with chlorine.
5. The article highlighted the decrease in sheet resistance with extended growth time for graphene glass fiber fabric. What is the underlying mechanism explaining the variation in sheet resistance as a function of growth time?

Reviewer #3 (Remarks to the Author):

In this article, the authors report a method for direct CVD growth of graphene on dielectric or insulating substrates using a high electronegative material (dichloromethane). Through the advantages of this material, the author presented the theoretical basis for the rapid growth mechanism and applications utilizing it. Specifically, the use of dichloromethane as a precursor accelerates multiple steps in the graphene growth process, leading to a dramatic increase in growth rate on non-metallic substrates like glass fiber fabric. This graphene-coated material, called graphene glass fiber fabric (GGFF), offers excellent electrical conductivity and is used to create highly sensitive and portable pressure sensors for applications in human motion and physiological signal monitoring.

The growth mechanism of graphene using dichloromethane, as presented by the authors, appears to be a novel approach. However, additional investigation is needed to substantiate its advantages. Particularly, specific analysis and in-depth discussion of various factors in the growth rate of graphene as a material are needed. In addition, their graphene-based conductive fiber material, which has the advantage of large area, require advanced demonstrations that go beyond those previously reported such as small patch-type ones for detecting pulse waves and sound waves. To publish this article for Nature Communication, it is judged that the major issues described above must be resolved.

Major comments

Further investigation and discussion regarding the growth rate of graphene are needed. The authors have defined the growth rate (v , min⁻¹) in terms of graphene coverage (θ) and growth time (t) using an equation $v = \theta/t$, which seems to have neglected several crucial variables in the context of real graphene growth. Therefore, in order to achieve a more precise and focused understanding of graphene growth, it is essential to conduct in-depth analyses of various critical parameters, such as activation energy, threshold barrier, precursor formation energy, and more [ref; ACS Nano, 2021, 15.4:7399-7408, reference, Science, 2013, 342.6159: 720-723.].

The author's argument regarding "the same growth rate" requires additional data for validation. While the author mentions increasing the amount of methane to achieve the same growth rate as dichloromethane, they do not provide specific results regarding the difference in growth rates with respect to the increase in methane. This omission could potentially lead to confusion among readers. Therefore, it is essential to supplement the argument with data that demonstrates how varying the amount of methane affects the growth rate, to substantiate the claim of achieving the same growth rate when using dichloromethane and methane.

Their graphene-based conductive fiber material, which has the advantage of large area, require application technologies that go beyond those previously reported such as small patch for detecting pulse waves and sound waves. Highly sensitive strain sensors that can be attached to the skin in a small area have already been reported using various materials (e.g., fibers, conductive polymers, and composites).

Minor comments

In the supplementary Figure 3, it is evident that the highest growth rate is observed when the H₂/precursor ratio is set at 5 compared to other ratios. This suggests that using a ratio of 5 for the precursor appears to be more effective. However, it appears that the author chose the ratio of 8. It is imperative that the author provides clear justification and explanation for this choice.

It would be good to show more details, such as a temperature profile diagram of the CVD process along with the flow rate of the precursors in the experiments performed by the author. (Journal of Materials Science, 2020, 55: 545-564., Science, 2013, 342.6159: 720-723.)

Demonstrating the application shown in Figure 4(e) through a live video would be more effective in highlighting to the reader the performance of the fabricated device and demonstrating its higher reliability.

Authors should indicate information about bending angles in Supplementary Figure 12

A description should be added of which samples were used for analysis (Figure 2 (d), (e) and Figure 4 pressure sensor application).

I recommend that the authors present GGFF fabricated from methane precursors as shown in Figure 1(b).

There is a typo in the TEM image introduction on page 10. Figure 3 should be changed to Figure 2. And there is a spacing error in the second line 'constructed in' in the Figure 4 caption.

RESPONSE TO REVIEWERS' COMMENTS

Reply to Reviewer #1

Wang et al. reported a method for directly growing graphene via CVD on glass fiber fabric. They employed the dichloromethane as a precursor to accelerate the graphene growth. The authors claimed that this approach increased the growth rate by approximately three orders of magnitude compared to the traditional methane precursor, significantly enhancing the graphene growth speed. In recent years, considerable research has been published on growing graphene on insulating substrates with different precursors. The use of dichloromethane in this work could be considered one of the optimizations for current growth techniques.

At its current state, frankly, I cannot recommend the publication due to numerous serious concerns. I would be willing to reconsider my recommendation if the authors can address all my questions and make considerable improvements. Refer to the following suggestions.

Our reply:

We would like to express our sincere appreciation to the reviewer for taking the time to thoroughly assess our manuscript and for providing valuable feedback and suggestions. We acknowledge the reviewer's concerns and recommendations, and we have tried addressing each of these issues and making considerable improvements to our research.

Comment 1:

Dichloromethane.

The authors claim, "the dichloromethane, a widely used organic solvent in industry, was first used as the carbon precursor for graphene CVD growth." It is not true. A quick online

search would reveal related reports on dichloromethane's effect on graphene growth, such as one work by Prof. Ding et al. in *Advanced Science*, 2022, 9(15): 2200737. This claim requires reevaluation.

Our reply:

Thank you very much for your recommendation of Prof. Ding et al.'s work (*Advanced Science* 2022, 9, 2200737), and this work indeed introduced dichloromethane into the graphene CVD growth system. To ensure a more accurate claim, according to the reviewer's suggestion, we have revised the corresponding description as follows: "*Importantly, to overcome the growth rate limit for graphene on the noncatalytic GFF, the dichloromethane, a widely used organic solvent in industry, was used as the carbon precursor for graphene CVD growth to increase the growth rate through accelerating the multiple CVD elementary steps, instead of one single elementary step as commonly reported in other precursor-modulating systems*" in the revised manuscript (Line 77–80, Page 4). **To make the citation more comprehensive, we have also cited the Prof. Ding et al.'s work as the Reference #50.** However, after our in-depth analyses, there are indeed some distinctions between our approach and Prof. Ding et al.'s

In the work by Prof. Ding et al., the mixed carbon precursor composed of chloroform (CHCl_3) and methanol was used as the carbon precursor for graphene growth, where methanol worked as the main carbon precursor, and chloroform (CHCl_3) mainly worked as an accelerator to promote the decomposition of methanol. In this work, they indeed briefly demonstrated the use of dichloromethane (CHCl_2) as the accelerator in the mixed carbon precursor, which is just for the comparisons of growth rate with CHCl_3 accelerator, and did not delve into the growth behaviors of dichloromethane-grown graphene in depth. Furthermore, it is important to note that Prof. Ding et al.'s study was based on the plasma-enhanced CVD (PECVD) route to synthesize vertical graphene. PECVD involves the energy coupling of a plasma generator to boost the dissociation of precursors,

resulting in a significantly different growth mechanism from the conventional thermal CVD system.

In contrast, in our work, dichloromethane is applied as an independent carbon precursor in a conventional thermal CVD system, and we extensively investigated the growth behaviors and mechanisms of graphene using dichloromethane as the carbon precursor. Our findings demonstrate that dichloromethane can facilitate a series of primitive steps in graphene CVD growth. It has a low decomposition energy barrier to promote the production of active carbon species, and more importantly, the produced high-electronegativity Cl radical can enhance the adsorption of active carbon species by Cl-CH₂ co-adsorption and facilitate H detachment from graphene edges. We hope this clarifies the differences in our approach and highlights the unique aspects of our research.

Comment 2):

The authors conducted similar DFT calculations to the work mentioned above. Their calculation results also suggest that CHCl₃ and CCl₄ could be better choices than CH₂Cl₂. The authors should comprehensively compare these choices and explain the advantages of using dichloromethane.

Our reply:

Thank you for the reviewer's valuable comment regarding the choice of carbon precursors for graphene CVD growth. According to the reviewer's advice, we conducted a comprehensive comparison and provided the detailed explanations for the reasons that we chose dichloromethane as the precursor.

As illustrated in the main text, the choice of carbon precursors containing high-electronegativity elements is an effective strategy for accelerating the CVD growth rate of graphene. As the reviewer pointed, CHCl₃ and CCl₄ may also be good choices for the fast growth of CVD graphene considering that they all contain the high-electronegativity chlorine element. However, it

should be noted that in the field of graphene growth, there is often a trade-off between the growth rate and crystallization quality of obtained graphene. For example, during the growth of large single-crystal graphene with low density of defects, in order to reduce the defective grain boundaries in CVD graphene, the suppression of graphene nucleation is required, where a very low amount of carbon precursor supply is usually required, which, on the other hand, leads to a significantly reduced growth rate. For example, in the reported work of Prof. Liu et al., the time required for growing micrometer-size graphene grains on SiO₂/Si substrates can be even as long as 3 days (*Adv. Mater.* 2014, 26, 1348–1353). Obviously, there is always a dilemma between the high growth rate and high crystal quality.

The similar dilemma between growth rate and quality can be anticipated in our chloromethane CVD growth system. On one hand, dichloromethane possesses a lower decomposition energy barrier than that of methane, thus enabling a large amount of active carbon species in CVD growth system. The introduction of highly electronegative Cl radical also enhances the capacity for GFF substrate to capture active carbon species. Therefore, the growth processes of graphene on the substrate can be greatly accelerated. On the other hand, excess active carbon species together with their enhanced adsorption on GFF, in turn, leads to a high graphene nucleation density, which will further bring more defective grain boundaries into as-grown graphene, and cause the crystal quality decrease of the graphene films. These were also discussed in the above-mentioned Prof. Ding et al.'s work (*Advanced Science* 2022, 9, 2200737), where they found the introduction of chlorine could promote the vertical graphene growth in PECVD, and yet excess chlorine atoms could suppress the growth quality of vertical graphene due to the excess active carbon species produced in the growth system.

To further clarify the effects of chlorine amount in carbon precursors on graphene CVD growth, we supplemented new experiments using chloroform (CHCl₃) as the precursor to fabricate

GGFF. We collected the Raman spectra of CHCl_3 -grown graphene for different growth time, as shown in **Response Fig. 1a**. Meanwhile, Raman spectra of CH_2Cl_2 -grown graphene in **Fig. 1c** in the main manuscript is also presented here as **Response Fig. 1b** for the comparison. It can be noticed that in the CHCl_3 system, the monolayer graphene film could be formed within ~ 5 s, which increased by ~ 6 times compared with that of dichloromethane (CH_2Cl_2). In return, the Raman I_D/I_G ratio of CHCl_3 -grown monolayer graphene obtained within the growth time of ~ 5 s is ~ 1.8 , obviously higher than that of CH_2Cl_2 -grown monolayer graphene (~ 1.4), which revealed the lower crystal quality of CHCl_3 -grown monolayer graphene. As the reviewer pointed, CCl_4 can be expected to realize a much higher growth rate than that of CH_2Cl_2 and CHCl_3 due to a higher ratio of the chlorine element, but according to the above analyses, the crystallization quality of CCl_4 -grown graphene will be further impaired. In summary, although CHCl_3 and CCl_4 have the expected potentials to achieve a higher growth rate of graphene, they will cause greater sacrifice of graphene crystal quality. In contrast, CH_2Cl_2 can achieve a better tradeoff between growth rate and crystal quality, which is the main consideration we chose CH_2Cl_2 in our work.

Response Fig. 1 | Comparison of Raman spectra between (a) CHCl_3 - and (b) CH_2Cl_2 -grown graphene (normalized to G peak intensity). Growth parameters: growth temperature of 1100 °C, H_2 flow of 120 sccm, CHCl_3 or CH_2Cl_2 flow of 15 sccm.

Comment 3):

Experimental evidence.

The claim "the increase of graphene layer thickness, which are further supported by the increasing intensity ratios of 2D and G peaks (I_{2D}/I_G) in Raman characterizations in Fig. 1c." is incorrect. Refer to the work from Ruoff's group in Nature Nanotechnology volume 15, pages 289–295, 2020.

Our reply:

Great thanks for the reviewer's comment, and we are sorry for the careless mistake in the data description in Fig. 1c. **We have confirmed that the I_{2D}/I_G values in Fig. 1c indeed decreased with the increase of graphene layer thickness** and have carefully checked the manuscript to avoid such careless mistake.

We have revised the corresponding description as "*It can be observed that contrasts of GGFFs gradually became darker with growth time extending, indicating the increase of graphene layer thickness, which are further supported by the **decreasing** intensity ratios of 2D and G peaks (I_{2D}/I_G) in Raman characterizations in Fig. 1c" in the revised manuscript" (Line 138, Page 7). **To make the citation more comprehensive, we have also cited Prof. Ruoff et al.'s work as the Reference #36.***

Comment 4):

The lack of significant change in G-band intensity with substantial thickness increase is puzzling. Additionally, the 2D over G peak ratio appears not correct.

Our reply:

Thank you for your inquiry and valuable observations regarding our Raman characterizations. The normalization of the G peak intensity has been made to give a direct comparison of D and 2D-

band intensities between different samples. Therefore, negligible change in G-band intensity could be observed for Raman spectra in our manuscript.

To avoid puzzling, we have added the corresponding illustrations "*Raman spectra corresponding to GGFs in (b) (normalized to G peak intensity)*" in the caption of **Fig. 1c** in the revised manuscript (Line 126, Page 7).

We are sorry for the careless mistake made in the description of the ratios of 2D and G peaks (I_{2D}/I_G), and it indeed decreased with the increase of graphene layer thickness, and this corrected interpretation will be reflected in our revised manuscript. We appreciate the reviewer's diligence in helping us make these improvements.

Comment 5):

The excessive D-band intensity, even surpassing the G band, suggests poor graphene crystallinity. The decrease in D-band intensity as thickness increases needs detailed explanations.

Our reply:

We appreciate the reviewer's comment and insightful observation regarding the D-band intensity in our Raman characterizations. As the reviewer pointed out, the intensity of D band in Raman spectra is an important indicator for the crystal quality of graphene, and the high D band (**Fig. 1c**) in the CH₂Cl₂-grown graphene does indeed suggest a potential issue with graphene crystallinity.

Actually, the growth of high-quality graphene on non-catalytic nonmetallic substrates has always been a significant technological challenge in this field. One of the primary reasons for this challenge is the inherently low catalytic activity and slow diffusion of carbon fragments on dielectric/insulating materials. As a result, conventionally grown graphene on such substrates tends

to exhibit small domain sizes, high-density grain boundaries, multilayer nucleation, and various defects, all contributing to relatively poor graphene crystallinity. These characteristics have been extensively demonstrated in a series of reported works (*Nat. Mater.* 2022, 21, 740–747; *J. Am. Chem. Soc.* 2019, 141, 11004–11008; *Nano Lett.* 2011, 11, 3612). Consequently, the distinct observation of the Raman D-band is consistent when graphene is synthesized on non-metal substrates.

More importantly, achieving high crystallinity quality and high growth rates simultaneously is widely recognized as an exceptionally challenging task in this field (*Nano Res.* 2016, 9, 3048; *Nat. Commun.* 2013, 4, 2096; *Chem. Rev.* 2018, 118, 9281–9343). Specifically, to realize a high growth rate, a large amount of carbon precursor supply is usually required, accompanied by the enhancement of graphene nucleation. The crystallization quality of graphene will be damaged since more defective grain boundaries are inevitably brought into as-grown graphene. Notably, our work prioritized increasing the growth rate of graphene on nonmetallic noncatalytic glass fiber, and an exceptionally high growth rate has been realized through using the dichloromethane precursor, which can facilitate a series of primitive steps during graphene CVD growth, such as the decomposition of carbon precursors, adsorption and nucleation of active carbon species, and further growing coalescence of graphene domains. As a result, in the dichloromethane CVD growth system, a high nucleation level is inevitable, leading to the formation of rich defective grain boundaries. Therefore, in this work, the achievement of the high growth rate of graphene may have to be accompanied by a slight sacrifice for crystal quality to some extent.

As the reviewer pointed, the D-band intensity in the Raman spectra in **Fig. 1c** is indeed decreased as graphene thickness increases. During our experiments, the thickness or the layer number increase of graphene films was realized by extending the growth time of graphene at high temperature (~1100 °C) in the CVD system. This process can be regarded as a high-temperature

annealing process for the already formed graphene layers, which can effectively repair the defects and restore graphitic structures in graphene films (*Chem. Mater.* 2017, 29, 7808–7815; *Adv. Mater.* 2011, 23, 1675–1678; *J. Mater. Sci. Technol.* 2015, 31, 599–606). As a result, the crystallization quality of graphene was gradually improved with growth time extending, presenting a decrease downward trend in D-band intensity as thickness increases.

Comment 6):

Additional characterization.

Raman maps are required on various thicknesses of graphene to assess layer uniformity.

Our reply:

Many thanks for the reviewer's valuable suggestion regarding the need for Raman maps on various thicknesses of graphene to assess layer uniformity. According to the reviewer's advice, we supplemented Raman mappings of GGFF that cover 3 different graphene thicknesses of 2 nm (**Response Fig. 2a**), 15 nm (**Response Fig. 2b**), 45 nm (**Response Fig. 2c**) in our research. The results show that graphene with varying layer thicknesses all possesses a satisfactory uniformity of Raman 2D-band intensity, which provide a visual representation of layer uniformity.

According to the reviewer's comment, we have added **Response Fig. 2** in the Supporting Information as **Supplementary Fig. 3**, and added the corresponding illustration "*As revealed in Supplementary Fig. 3, I_{2D} mappings of the fabricated GGFF with varying graphene thicknesses provided the visual representations of high layer uniformity*" in the revised manuscript (Line 141–143, Page 8).

Response Fig. 2 | Intensity of 2D peak mappings of the fabricated GGFF with graphene thickness of (a) ~2 nm, (b) ~15 nm, and (c) ~45 nm. The 2D peak intensity of graphene ribbons in (b, c) was normalized by that of graphene ribbon in (a).

Comment 7):

The authors should provide a detailed characterization of the graphene ribbon in Figure 1D. Better, including the EDX mapping in SEM characterization. The cross-sectional TEM of grown graphene is necessary to confirm their layered structure.

AFM image is confusing in Fig. 1b. Ensure accurate line profiles and provide AFM characterization for various thicknesses of graphene. Indicate the graphene region.

Our reply:

We appreciate the reviewer's constructive advice for providing more detailed characterizations of the graphene ribbon Fig. 1d. According to the reviewer's suggestion, we have supplemented the EDX mapping, cross-sectional TEM, and AFM of graphene layers with different thicknesses in our revised manuscript:

(1) EDX mapping in SEM characterization:

As displayed in scanning electron microscopy (SEM) image in Fig. 1d, the graphene ribbon was obtained by etching the GFF core in hydrofluoric acid and graphene layers collapsing onto the silicon substrate. We have incorporated EDX (Energy-Dispersive X-ray) mapping of C element in

the SEM characterization to offer insights into the elemental composition and distribution in the graphene ribbon (**Response Fig. 3**). In **Response Fig. 3**, the region between the two white dashed lines was detected with obviously stronger intensity of C element than that of exterior region. Therefore, the graphene ribbon can be proved to be located at the region between the two white dashed lines.

According to the reviewer's comment, we have added **Response Fig. 3** in the Supporting Information as **Supplementary Fig. 5**, and added the corresponding illustration "*The graphene ribbon was obtained by etching the GFF core in hydrofluoric acid and graphene layers collapsing onto the silicon substrate, as displayed in scanning electron microscopy (SEM) image in Fig. 1d, and the corresponding energy-dispersive X-ray (EDX) mapping of the ribbon was shown in Supplementary Fig. 5*" in the revised manuscript (Line 145–148, Page 8).

Response Fig. 3 | EDX mapping of C element for the corresponding graphene ribbon in Fig. 1d (transferred onto the silicon substrate).

(2) Cross-sectional TEM of grown graphene:

We acknowledge the importance of confirming the layered structure of the grown graphene. To address this, we have supplemented the cross-sectional transmission electron microscopy (TEM)

images of graphene grown on GGF substrates with four different layer thicknesses (**Response Fig. 4**) allowing us to verify the layered structure of the graphene.

According to the reviewer's comment, we have added **Response Fig. 4** in the Supporting Information as **Supplementary Fig. 4** and added the corresponding illustration "*Cross-sectional transmission electron microscopy (TEM) images of GGFF further verified the nice layered structure of as-grown graphene (Supplementary Fig. 4)*" in the revised manuscript (Line 143–145, Page 8).

Response Fig. 4 | Transmission electron microscopy (TEM) images of graphene grown on glass fiber with different layer thicknesses. Growth parameters: growth temperature of ~ 1100 °C, H₂ flow of 120 sccm, CH₂Cl₂ flow of 15 sccm, growth time of (a) ~ 30 s, (b) 40 s, (c) 1 min, and (d) 2 min. (a) was the TEM image of monolayer graphene transferred onto a TEM grid. (b–d) Cross-sectional TEM images of GGFF samples prepared by focused ion beam (FIB) techniques with chromium (Cr) layers deposited to protect samples from possible damage during sample preparation.

(3) AFM characterization:

According to the reviewer's advice to ensure the accuracy of the line profile in **Fig. 1e**, we remeasured the line profile at a different region in the AFM image, as represented in **Response Fig. 5**, and replaced **Fig. 1e** by **Response Fig. 5** and revised the corresponding caption "*AFM image and height profile of the graphene ribbon along the marked dashed orange line, showing ribbon thickness of ~2 nm (twice the thickness of the grown graphene films), corresponding to 1–2 layers of graphene*" in the revised manuscript (Line 128–130, Page 7). We hope these modifications contribute to a more clear and accurate understanding of our findings.

Response Fig. 5 | AFM image and height profile of the graphene ribbon along the marked dashed orange line, showing ribbon thickness of ~2 nm (twice the thickness of the grown graphene films), corresponding to 1–2 layers of graphene.

(4) AFM characterizations for various thicknesses of graphene:

According to the reviewer's advice, we also supplemented more data about AFM characterizations for graphene with various thicknesses. We achieved the preparations of GGFF with graphene layers of varying thickness through the modulation of growth time. Subsequently, the graphene ribbons with different graphene thicknesses were obtained by etching the GFF core in hydrofluoric acid and graphene layers collapsing onto the silicon substrate. In this way, the thickness of graphene films can be directly measured by AFM characterizations. As presented in **Response Fig. 6**, it can be noticed that the graphene layers with thickness in a wide range can be

obtained through growing conditions modulating.

According to the reviewer's comment, we have added **Response Fig. 6** in the Supporting Information as **Supplementary Fig. 6**, and added the corresponding illustration "*Moreover, the AFM characterizations of graphene ribbons with different thicknesses obtained through growth time modulation were also provided in Supplementary Fig. 6, suggesting a good capacity for layer thickness control in our dichloromethane CVD growth system*" in the revised manuscript (Line 151–154, Page 8).

Response Fig. 6 | AFM characterizations of graphene ribbons with different thickness obtained through growth time modulation during graphene CVD growth process.

Comment 8):

Comparison

The claim about the high energy consumption due to CVD growth temperature contradicts the use of 1100°C growth temperature in this work. Explain this.

Our reply:

We appreciate your comment and your observation regarding the high energy consumption due to CVD growth temperature. It is true that high-temperature CVD process (usually >1000°C) will bring high energy consumption. According to our rough estimation, the CVD furnace (Thermo Scientific) we used can consume about 5.1 kW electric energy when it works at heating temperature of ~1100°C. The longer the working time is, the higher energy consumption will be costed. In our work, our approach is to use dichloromethane precursor to accelerate graphene CVD growth and shorten the growth time consumed, so as to save the electric energy consumed by the CVD furnace and relieve the issue of high energy consumption. To be more quantitatively, according to the growth data in **Fig. 2b**, in dichloromethane system, it only took ~0.5 min to realize the full-coverage of continuous graphene films with the energy consumption during this growth process of ~0.0425 kW h, while ~480 min was needed in conventional methane system to realize the same state, with the energy consumption of ~40.8 kW h. Therefore, the use of dichloromethane as a precursor, along with growth optimizations, allowed us to achieve ~99.9% energy saving, which will effectively mitigate the energy consumption issue in the high-temperature graphene CVD growth.

Comment 9):

The growth rate comparison seems inappropriate and meaningless, as it compares the rate of multilayer graphene with studies on monolayer graphene growth. In most cases of single-layer graphene growth, the primary goal is precisely controlling the layer thickness and uniformity. Refer to the work in *Advanced Materials*, 2019, 31(35): 1903615. Again,

references 17, 18, 24, and 40 pertain to the growth of monolayer graphene. The authors are suggested to focus on comparing mass-produced multilayer graphene growth, whether on metals or glasses.

Our reply:

We genuinely appreciate the valuable feedback from the reviewer regarding the comparison of growth rates. Actually, we have made a mistake in the reference numbering in the original **Fig. 2c**, that during the draft revision, some adjustments were made to the reference list of the whole manuscript, but the reference numbering in **Fig. 2c** was not updated accordingly. Therefore, the references cited in the original **Fig. 2c** looked very inappropriate. In light of this, the corresponding and correct literatures cited in original **Fig. 2c** was listed below, and the corresponding figure is now presented as **Response Fig. 7**.

17. Wang, H. et al. Primary nucleation-dominated chemical vapor deposition growth for uniform graphene monolayers on dielectric substrate. *J. Am. Chem. Soc.* **141**, 11004–11008 (2019).
21. Chen, J. et al. Oxygen-aided synthesis of polycrystalline graphene on silicon dioxide substrates. *J. Am. Chem. Soc.* **133**, 17548–17551 (2011).
25. Chen, X. D. et al. Fast growth and broad applications of 25-inch uniform graphene glass. *Adv. Mater.* **29**, 1603428 (2017).
36. Xie, H. et al. H₂O-etchant-promoted synthesis of high-quality graphene on glass and its application in see-through thermochromic displays. *Small* **16**, 1905485 (2020).
38. Liu, R. et al. CO₂-promoted transfer-free growth of conformal graphene. *Nano Res.* **16**, 6334–6342 (2022).
39. Chen, Y. et al. Growing uniform graphene disks and films on molten glass for heating devices and cell culture. *Adv. Mater.* **27**, 7839–7846 (2015).
40. Wei, S. et al. Water-assisted rapid growth of monolayer graphene films on SiO₂/Si substrates. *Carbon* **148**, 241–248 (2019).
41. Chen, Z. et al. Fast and uniform growth of graphene glass using confined-flow chemical vapor deposition and its unique applications. *Nano Res.* **9**, 3048–3055 (2016).
42. Xie, Y. et al. Ultrafast catalyst-free graphene growth on glass assisted by local fluorine supply. *ACS Nano* **13**, 10272–10278 (2019).

Response Fig. 7 | Comparisons between the growth rates of graphene obtained in this work with those reported in literature.

In addition, according to the reviewer’s advice, we focused on comparing growth rates of multilayer graphene, and reevaluated all the references in **Response Fig. 7**. After reassessment, we recognized that the original **References #17, #36, and #38** (*J. Am. Chem. Soc.* 2019, 141, 11004–11008; *Small* 2020, 16, 1905485; *Nano Res.* 2022, 16, 6334–6342) may be inappropriate for the comparisons, which have been removed in the revised **Fig. 2c** (also **Response Fig. 8**).

Ni and Cu–Ni alloy emerge as the most prevalent choices for graphene synthesis owing to their excellent catalytic capability for graphene CVD growth, cost-effectiveness, and widespread availability. Both of them tend to yield multilayer graphene. For example, the work of Prof. Zhou (*J. Phys. Chem. Lett.* 2010, 1, 20, 3101–3107) outlined the multilayer graphene growth on Ni substrate, which had a growth rate of 0.1 min⁻¹. In addition, in the reported work of Prof. Rack et al. (*Nanomaterials* 2022, 12, 1553), the use of Cu–Ni alloy for graphene growth with adjustable layer thickness has been investigated, where graphene was synthesized on Ni–Cu with a growth rate of 0.03 min⁻¹. According to the reviewer’s advice to include the multilayer graphene growth on metals for more comprehensive comparisons, we have included the two above references as **References #42 and #43** in the revised growth rate comparison in **Fig. 2c** (also **Response Fig. 8**),

and added the corresponding illustration "To the best of our knowledge, the graphene growth rate obtained in this work is much higher than that in reported literatures for graphene grown on nonmetallic substrates, *even on the metal substrates such as Ni and Ni–Cu alloy*" in the revised manuscript (Line 195–198, Page 10). The corresponding references in **Response Fig. 8** (revised Fig. 2c) are also listed as follows.

Response Fig. 8 | Comparisons between the growth rates of graphene obtained in this work with that reported.

21. Chen, J. et al. Oxygen-aided synthesis of polycrystalline graphene on silicon dioxide substrates. *J. Am. Chem. Soc.* **133**, 17548–17551 (2011).
25. Chen, X. et al. Fast growth and broad applications of 25-inch uniform graphene glass. *Adv. Mater.* **29**, 1603428 (2017).
39. Chen, Y. B. et al. Growing uniform graphene disks and films on molten glass for heating devices and cell culture. *Adv. Mater.* **27**, 7839–7846 (2015).
40. Wei, S. et al. Water-assisted rapid growth of monolayer graphene films on SiO₂/Si substrates. *Carbon* **148**, 241–248 (2019).
41. Chen, Z. et al. Fast and uniform growth of graphene glass using confined-flow chemical vapor deposition and its unique applications. *Nano Res.* **9**, 3048–3055 (2016).
42. Khanna, S. R., Stanford, M. G., Vlasiouk, I. V. & Rack, P. D. Combinatorial Cu-Ni alloy thin-film catalysts for layer number control in chemical vapor-deposited graphene. *Nanomaterials* **12**, 1553 (2022).
43. Zhang, Y. et al. Comparison of graphene growth on single-crystalline and polycrystalline Ni by chemical vapor deposition. *J. Phys. Chem. Lett.* **1**, 3101–3107 (2009).
44. Xie, Y. et al. Ultrafast Catalyst-Free Graphene Growth on Glass Assisted by Local Fluorine Supply. *ACS Nano* **13**, 10272–10278 (2019).

Comment 10):

Growth mechanism.

The authors should provide detailed growth conditions, including temperature, chamber pressure, gas flows, and time, on various experiments for reproducibility.

Our reply:

Thank the reviewer for the suggestion to provide clearer descriptions for the samples used for analyses in specific figures. In our revised manuscript, we have first supplemented the more detailed descriptions about the experiment method in Method section, and added the corresponding illustrations in the revised manuscript to specify which samples were utilized for analysis in different experiments.

Methods

Dichloromethane precursor-based CVD growth of GGFF

Commercially available GFF of ~0.1 mm thickness (Wuhan Sino Type Optoelectronic Technology CO., LTD) was first annealed under ~500 °C in ambient air for 2 h to remove the coated polymer (same treatment for below). After being carefully cleaned, GFF was placed at the center of a high-temperature furnace with the low-pressure CVD (LPCVD) system. In a typical procedure, the system was evacuated to a base pressure of <1 Pa and heated to 1100 °C under a H₂ flow of 120 sccm. Subsequently, dichloromethane vapor with desired flow (5–24 sccm) was pumped to the chamber for the graphene growth. Throughout the growth process, the chamber pressure was held at approximately 400–500 Pa depending on the flow of dichloromethane vapor. The growth of graphene lasted for 0.5–10 min, followed by natural cooling process to room temperature under a H₂ flow of 20 sccm and Ar flow of 200 sccm. The temperature profile diagrams of the CVD process and information about the flow rate of precursors were shown in Supplementary Fig. 1a^{64,65}. GGFF samples for sheet resistance mapping (Fig. 1f) and TEM characterization (Fig. 2d) were grown for ~10 min and 0.5 min, respectively, with the H₂ flow of 120 sccm and the dichloromethane vapor flow of 15 sccm at ~1100 °C.

Methane precursor-based CVD growth of GGFF under consistent carbon supplies with dichloromethane-based growth

The same GFF substrates and LPCVD method were used in experiments. The system was evacuated to a base pressure of <1 Pa and heated to ~1100 °C under a H₂ flow of 120 sccm. Subsequently, methane gas with desired flow (5–24 sccm) was introduced to the chamber for the graphene growth. Throughout the growth process, the chamber pressure was held at approximately 400–500 Pa depending on the flow of methane. In each individual comparative experiment, the methane flow remained consistent with the flow of dichloromethane vapor to ensure consistency across the comparative experiments. The growth of graphene lasted for 8–15 h, followed by natural cooling process to room temperature under a H₂ flow of 20 sccm and Ar flow of 200 sccm. The temperature profile diagrams of the CVD process and information about the flow rate of precursors were shown in Supplementary Fig. 1b^{64,65}.

Methane precursor-based CVD growth of GGFF under consistent growth rate with dichloromethane-based growth

To obtain the same growth rate of methane-grown graphene (Fig. 2e) with that of dichloromethane-grown graphene, the H₂/CH₄ ratio was further reduced to 1:5 with the H₂ flow of 10 sccm and the methane flow of 50 sccm. The growth temperature was maintained at ~1100 °C and the CVD growth process lasted for ~0.5 min.

Fabrication of GGFF flexible pressure sensors

GGFF for the pressure sensor fabrication was synthesized with the H₂ flow of 120 sccm and the methane flow of 15 sccm at 1100 °C for ~4 min. Subsequently, copper double-sided tapes were stuck on the both ends of GGFF to serve as the electrodes. Also, copper wires were connected to make an electrical connection. Finally, polypropylene films tightened by a hot plate were used to package the sensor.

In addition, we have also checked the entire text, and added more necessary information for the samples illustrated in the manuscript, which are also listed below.

(1) *Figure 2d, e* presents the transmission electron microscope (TEM) images of dichloromethane- and methane-grown graphene obtained at the same growth rate ($\sim 2 \text{ min}^{-1}$) (see more details in Supplementary Fig. 9 and Methods section ‘Dichloromethane precursor-based CVD growth of GGFF’ and ‘Methane precursor-based CVD growth of GGFF under consistent growth rate with dichloromethane-based growth’), which reveal the longer-range crystalline order of dichloromethane-grown graphene in contrast to the amorphous structure of the methane-grown graphene. (Line 207–213, Page 11)

(2) As shown in Supplementary Fig. 18, a flexible sensor was fabricated based on GGFF encapsulated with the polypropylene films (see more details in Methods section ‘Fabrication of GGFF flexible pressure sensors’). (Line 326–329, Page 17)

(3) The preparation process of GGFF through the direct CVD graphene growth on GGF is schematically presented in Fig. 1a, where dichloromethane was creatively introduced into the CVD system as the carbon precursor (see more details in Supplementary Fig. 1a and Methods section ‘Dichloromethane precursor-based CVD growth of GGFF’). (Line 107–110, Page 6)

(4) The sheet resistance mapping on GGFF in Fig. 1f reveals the high uniformity of the electrical conductivity with a mean value of $35.0 \pm 2.3 \Omega \text{ sq}^{-1}$ over the $5 \text{ cm} \times 5 \text{ cm}$ area (see more details in Methods section ‘Dichloromethane precursor-based CVD growth of GGFF’). (Line 155–158, Page 8)

(5) The sheet resistance of GGFF as a function of growth time obtained with dichloromethane and methane precursors was systematically compared in Fig. 2b (carbon supplies introduced into the two CVD systems remained consistent) (see more details in Methods section ‘Dichloromethane

precursor-based CVD growth of GGFF' and 'Methane precursor-based CVD growth of GGFF under consistent carbon supplies with dichloromethane-based growth'). (Line 180–184, Page 10)

Comment 11):

The mechanism of graphene nucleation on fibers is required to be explained. Show individual graphene grains on fibers in SEM images.

Our reply:

Thank you for your valuable feedback and the request for a more detailed explanation of the mechanism of graphene nucleation on fibers in our research, and according the reviewer's advice, we have supplemented the SEM (**Response Fig. 9**) and AFM (**Response Fig. 10**) images of GGFF with different graphene coverage, which showed the individual graphene grains on fiber.

To visually illustrate graphene nucleation on fibers, we have incorporated SEM images at various growth stages (**Response Fig. 9**). These images revealed the gradual growth process of the continuous graphene films from the low-coverage small graphene domains. However, due to the non-conducting property of GFF substrates even with isolated graphene domains (period I in **Fig. 2a**) and the resolution limitation of SEM, the observed graphene nuclei at low coverage are very unclear. To address this, we have included AFM images with higher resolution for the observation of graphene grains (**Response Fig. 10**). **Response Fig. 10** distinctly displays individual graphene grains on the fiber. At a coverage stage of ~3%, the average domain size of graphene is ~40 nm. As the growth time extends, the average domain size increases to ~50 nm, and the coverage increases to ~44%. Upon reaching a coverage of ~76%, the average domain size reaches ~70 nm, and the fusion and junctions between nuclei were observed. Therefore, the growth of graphene on glass fiber involves several typical processes of graphene CVD growth, i.e., graphene nucleation, domain growth, and coalescence.

Response Fig. 9 | SEM images of graphene on glass fiber substrate with different coverage.

Response Fig. 10 | AFM images of graphene on glass fiber substrate with different coverage.

Comment 12):

Errors.

Numerous typos and errors in the text need correction. Wrong Figure number "Figure 3d, e."

Our reply:

Thank you for pointing out the typos and errors in the text, as well as the incorrect Figure number reference. We have made the corrections for all typos throughout the main manuscript and revised the wrong figure number "**Fig. 3d, e**" to "**Fig. 2d, e**" in the sentence "*Figure 2d, e presents the transmission electron microscope (TEM) images of dichloromethane- and methane-grown graphene obtained at the same growth rate ($\sim 2 \text{ min}^{-1}$) (see more details in Supplementary Fig. 9 and Methods section 'Dichloromethane precursor-based CVD growth of GGFF' and 'Methane precursor-based CVD growth of GGFF under consistent growth rate with dichloromethane-based growth'), which reveal the longer-range crystalline order of dichloromethane-grown graphene in contrast to the amorphous structure of the methane-grown graphene*" in the revised manuscript (Line 207–213, Page 11). In our revised version, we have diligently reviewed the entire text to improve the quality and accuracy of our manuscript.

Reply to Reviewer #2

The article submitted by Kun Wang et al. presents an intriguing study on the direct chemical vapor deposition (CVD) growth of graphene on dielectric/insulating materials, particularly focusing on glass fiber fabric as the substrate. The novel approach of using dichloromethane as the carbon precursor to promote and accelerate graphene growth on the noncatalytic nonmetallic substrates will be a promising route to break the bottleneck for graphene mass production and practical applications. The mechanisms behind this choice, particularly the low decomposition energy barrier of dichloromethane and its role in enhancing carbon species adsorption and hydrogen detachment at graphene edges, are explained clearly and logically. I believe this work will attract wide interests from the areas of graphene and 2D materials in the academia, as well as the attentions from the industry, who are concerned about the practical applications of graphene. Therefore, I recommend it can be accepted after a minor revision emphasizing the following questions.

Our reply:

Thank the reviewer for the positive and encouraging feedback on our manuscript. We appreciate the reviewer's recognition of the novelty and potential impact of our study on the direct CVD growth of graphene using dichloromethane as a carbon precursor on dielectric/insulating materials, particularly glass fiber fabric. The reviewer's comments regarding the clear and logical explanation of the mechanisms behind the choice of dichloromethane and its role in the graphene growth process are valued. We have carefully addressed the questions and suggestions the reviewer emphasized during the revision process, aiming to provide a more comprehensive and refined manuscript to meet the standards for acceptance.

Comment 1):

The authors demonstrated the rapid growth of graphene in a dichloromethane system to achieve high electrical conductivity. It will be better to provide more detailed application scenarios where the conductive graphene glass fiber fabric is advantageous.

Our reply:

Thanks for the reviewer's valuable advice for providing more detailed application scenarios for the conductive graphene glass fiber fabric. Our fabricated conductive GGFF has shown great potentials in various applications such as electromagnetic interference (EMI) shielding, electric heating as well as radiant heating.

For example, the unique woven structure composed of warp and weft yarns (each containing thousands of fibers) in GGFF constructed the characteristic hierarchical conductive configuration, which brought about distinctive advantages for applications in EMI shielding. When electromagnetic wave (EMW) was incident on GGFF, part of it was reflected due to the interactions with the surface free charge carriers of graphene, and the reflection loss increases with the material's conductivity. Then, the residual EMW entered the inside of the weaving structure of GGFF, where it was multi-reflected between multiple adjacent fibers and the multi-absorption was accompanied. Consequently, the EMW loss can be largely enhanced. More specifically, when EMW is incident on the single graphene glass fiber in the fabric, it will be reflected by graphene layers, meanwhile, part of it will be absorbed through the conductivity loss and polarization relaxation loss in graphene layers. In addition, the incident EMW can induce the uneven charge distribution at the interface between graphene and glass fiber, which will cause the interface polarization to enhance the EMW loss. Moreover, the conductive GGFF also has many other advantages such as lightweight, flexibility and high strength, which can better meet the material requirements of industrial EMI shielding applications. Actually, the application of GGFF in EMI shielding has been explored in our previous work, which has been reported as a flexible EMI

shielding material with ultrahigh shielding effectiveness (SE) ($\sim 10^7$ dB across a broad frequency range (1–18 GHz) (*Adv. Mater.* 2022, 34, 2202982).

Benefiting from the excellent conductivity, lightweight, flexibility, and high strength, GGFF shows promising potentials for the electrothermal applications, as shown in **Response Fig. 11** (*Small Methods* 2022, 6, 2200499). The Joule heater constructed based on GGFF showed the uniform heating temperature, fast electrothermal response and high heating rate (for example, under the input power density of 5000 W m^{-2} , the response time and heating rate of GGFF-heater was 6 s and $29.7 \text{ }^\circ\text{C s}^{-1}$, respectively). In addition, the high flexibility and deformation robustness are also remarkable advantages of GGFF during practical applications. Therefore, GGFF showed the promising application potentials in the areas of electric heating. For example, the demand for efficient anti/de-icing solutions has surged in recent years, especially in critical sectors of aviation and renewable energy, for example, the anti/de-icing of aircraft and wind turbine blades. The application of GGFF in the electric heating anti/de-icing has been explored in our previous reported work, which exhibited exceptional anti/de-icing performances (*Small Methods* 2022, 6, 2200499).

Response Fig. 11 | Electrothermal performances and deicing ability of GGFF (*Small Methods* 2022, 6, 2200499). **a**, Infrared image of a 180°-bent GGFF-based heater with the heating temperature of $171.4 \pm 3.6 \text{ }^\circ\text{C}$ ($20 \text{ cm} \times 15 \text{ cm}$). **b**, Temperature profiles of $10 \text{ cm} \times 10 \text{ cm}$ -sized GGFF at various power densities.

Furthermore, conductive GGFF exhibits excellent performances in the field of radiation heating, thanks to the excellent infrared radiation ability of both graphene and glass fiber. Specifically, graphene has a wide spectral range and stable light absorption, and the infrared radiation of graphene materials can be effectively controlled by the number of graphene layers. Notably, glass fiber is also a good infrared radiation material originating from the Si–O bond vibration with high emissivity of around 0.8. In this way, GGFF possesses dual emitting elements, graphene and glass fiber, which greatly enhances the infrared radiation capability (with emissivity >0.9), especially when the graphene layer number is limited. The higher infrared emissivity makes GGFF more competitive in the field of infrared electric heating compared to metal electric heating wires, which typically have a low emissivity of <0.1. In addition, in our reported work, GGFF has been developed into a lightweight, flexible infrared electrothermal device, which presented high infrared emissivity, thermal radiation efficiency, fast electrothermal response, and uniform heating, being highly expected as an efficient and energy-saving radiant heating solution (*ACS Nano* 2022, 16, 2577–2584).

Comment 2):

The author claimed the high flexibility of the fabricated graphene glass fiber fabric, but didn't present the actual photos of the fabric being grasped, folded, twisted etc., and the morphology of graphene on the fiber after the mechanical deformations, is there any detachment or peeling happened? Suggest to provide more information.

Our reply:

We appreciate the reviewer's suggestion to provide a more comprehensive assessment of the flexibility of the fabricated GGFF. To address these concerns, we have conducted new experiments to impose the various mechanical deformations, such as grasping, folding, and twisting, on GGFF,

as shown in **Response Fig. 12a** (every deformation action was repeated for 100 times). **Response Fig. 12b–e** presents SEM images of GGFF after being grasped, folded, and twisted. As a result, the macro- and micro-morphologies of GGFF presented negligible change and no peeling of graphene layers was observed.

To provide a more thorough understanding of the flexibility and mechanical performances of GGFF, we have added **Response Fig. 12** in the Supporting Information as **Supplementary Fig. 20**, and added the corresponding illustration "*Applied as flexible sensors, the performance stability under various mechanical deformations was the significant premise. As shown in Supplementary Fig. 20, after repeated deformations of twisting, grasping, and folding, the morphology of GGFF presented negligible change and no peeling of graphene layers was observed, suggesting the high flexibility and interfacial stability of GGFF*" in the revised manuscript (Line 346–350, Page 18).

Response Fig. 12 | **a**, Photographs of GGFF being twisted, grasped, and folded. **b–e**, SEM images of (b) pristine GGFF and GGFF after being (c) grasped, (d) twisted, and (e) folded (every deformation action was repeated for 100 times). After the repeated mechanical deformations, the morphology of GGFF presented negligible change and no peeling of graphene layers was observed.

Comment 3):

The AFM image in Fig. 1e revealed an interlayer distance of graphene in the collapsed ribbon that appears relatively larger than the intrinsic value of graphene (~0.34 nm). Can the authors provide insights into the reasons behind this observed difference in interlayer distance?

Our reply:

We appreciate your comment regarding the interlayer distance of graphene in the collapsed ribbon as shown in Fig. 1e. The graphene bands shown in Fig. 1e were transferred onto silicon wafers through wet etching, and the interlayer distance of graphene was measured by AFM. The excessive interlayer distance of graphene measured may be influenced by both the etching transfer process and AFM testing.

Specifically, during the etching transfer process, graphene glass fibers are first soaked in a solution of hydrofluoric acid to remove the glass fiber substrate. The resulting unsupported graphene ribbons will float on the liquid surface, and then be transferred to the silicon wafer and washed multiple times with deionization to remove surface oxide particles and solvents. The flexibility and thinness of graphene make it susceptible to mechanical deformation in transfer processes (*Nat. Commun.* 2022, 13, 4409–4418; *Nat. Nanotechnol.* 2013, 8, 356–362), which may affect the accuracy of the interlayer distance measurements. The high sensitivity of graphene to water molecules and oxygen may lead to adsorption on the surface of graphene films, especially at defective points, resulting in graphene ribbon with high concentrations of p-type dopants (*Nano Lett.* 2012, 12, 2751–2756; *Nanoscale* 2020, 12, 10890–10911), which also lead to the interlayer expansion (*Adv. Energy Mater.* 2017, 7, 1602684).

Moreover, the measured height from AFM is dependent upon a number of interactions including tip-sample, sample-substrate, and tip-substrate. Another possible cause of the overestimation of graphene height by AFM is the adlayers between graphene and the substrate, which may create a buffer between graphene ribbon and substrate. In fact, particularly for single-layer graphene, reported values of interlayer distance ranging from 0.4 to 1.7 nm varies widely when compared to the interlayer distance of graphite (0.335 nm) (*Nat. Photonics* 2019, 13, 754–759; *J. Am. Chem. Soc.* 2011, 133, 2334–2337; *Sci. Rep.* 2014, 4, 6003–6008).

Comment 4):

Given that chlorine is produced by dichloromethane in the CVD system, there's a concern about potential chlorine doping of the as-grown graphene. Suggest investigating whether the as-grown graphene is indeed doped with chlorine.

Our reply:

Thank you for your insightful comment regarding the potential chlorine doping of the as-grown graphene. We acknowledge the importance of investigating this possibility to ensure the integrity of our results. In our revised manuscript, we added the X-ray photoelectron spectroscopy (XPS) characterization to assess the presence of chlorine in the graphene samples. In the survey XPS spectra (**Response Fig. 13a**) and the high-resolution of Cl element (**Response Fig. 13b**), no detectable signal peak of Cl element at around 200 eV was found, indicating that there was no Cl doping introduced into the graphene layers.

Response Fig. 13 | XPS survey spectra (a) and high-resolution XPS spectra of Cl (b) of as-fabricated GGFF.

Comment 5): The article highlighted the decrease in sheet resistance with extended growth time for graphene glass fiber fabric. What is the underlying mechanism explaining the

variation in sheet resistance as a function of growth time?

Our reply:

Thank you for the reviewer's comment regarding the variation in sheet resistance as a function of growth time for GGFF. The variation in sheet resistance with extended growth time is simultaneously influenced by the thickness and crystal quality of the graphene layers.

First, with the growth time being extended, the number of graphene layers was increased, as shown in the new supplemented data in **Response Fig. 14**, which showed the layer thickness as a function of growth time. Therefore, more transport channels of charge carriers are provided, resulting in the decrease of the sheet resistance for GGFF. This is one of the significant reasons for the decrease in sheet resistance with extended growth time. In addition, the growth time extending was accompanied by the increase of the high-temperature annealing time of graphene layers. During this process, large amounts of defects will be repaired, and the crystal quality of the graphene layers can be largely improved. This was confirmed by the decreased D-band in Raman spectrum of graphene as growth times extending (**Fig. 1c**), which can reflect the degree of crystal structure disorder. The defects repairing will largely weaken the carrier scattering and enhance the carrier mobility in graphene layers. Therefore, the crystal quality improvement was also a significant reason for the decrease in sheet resistance with the extension of growth time.

Response Fig. 14 | Graphene layer thickness as a function of growth time for GFFF prepared using dichloromethane and methane precursors.

Reply to Reviewer #3

In this article, the authors report a method for direct CVD growth of graphene on dielectric or insulating substrates using a high electronegative material (dichloromethane). Through the advantages of this material, the author presented the theoretical basis for the rapid growth mechanism and applications utilizing it. Specifically, the use of dichloromethane as a precursor accelerates multiple steps in the graphene growth process, leading to a dramatic increase in growth rate on non-metallic substrates like glass fiber fabric. This graphene-coated material, called graphene glass fiber fabric (GGFF), offers excellent electrical conductivity and is used to create highly sensitive and portable pressure sensors for applications in human motion and physiological signal monitoring.

The growth mechanism of graphene using dichloromethane, as presented by the authors, appears to be a novel approach. However, additional investigation is needed to substantiate its advantages. Particularly, specific analysis and in-depth discussion of various factors in the growth rate of graphene as a material are needed. In addition, their graphene-based conductive fiber material, which has the advantage of large area, require advanced demonstrations that go beyond those previously reported such as small patch-type ones for detecting pulse waves and sound waves. To publish this article for Nature Communication, it is judged that the major issues described above must be resolved.

Our reply:

In response to the reviewer's comments, we appreciate the constructive advice for providing the in-depth investigation to substantiate the advantages of graphene growth using dichloromethane. We have provided additional analyses and discussions on various factors affecting the graphene growth rate, including thermal and kinetic parameters, activation energy, threshold barriers, and electronic information. Furthermore, we have supplemented data to provide clearer comparison of

growth rate between dichloromethane and methane. Additionally, we have emphasized the versatility and potential applications of our GGFF, particularly in large-area scenarios, such as aircraft wing sensors and anti/de-icing solutions for aircraft and wind turbine blades. We have also included a live video demonstrating the performances of the fabricated GGFF sensor in the context of sound wave detection. Moreover, we have addressed other specific points raised by the reviewer, including providing a temperature profile diagram of the CVD process, details about bending angles, and information about the samples used in specific figures. Overall, we believe these modifications enhance the clarity, comprehensiveness, and reliability of our article, addressing the reviewer's concerns and providing a more robust foundation for potential publication in Nature Communications.

Comment 1:

Major comments

Further investigation and discussion regarding the growth rate of graphene are needed. The authors have defined the growth rate (v , min^{-1}) in terms of graphene coverage (θ) and growth time (t) using an equation $v = \theta/t$, which seems to have neglected several crucial variables in the context of real graphene growth. Therefore, in order to achieve a more precise and focused understanding of graphene growth, it is essential to conduct in-depth analyses of various critical parameters, such as activation energy, threshold barrier, precursor formation energy, and more [ref; ACS Nano, 2021, 15.4:7399-7408, reference, Science, 2013, 342.6159: 720-723].

Our reply:

We appreciate the reviewer's comment regarding the expression of graphene growth rate in our research. The expression of graphene growth rate with graphene coverage (θ)/growth time (t)

is a widely recognized and used approach in the field of graphene growth. For example, in the work by Dr. Geohegan et al. (*Carbon* 2014, 67, 417–423), they analyzed the kinetics of the graphene growth by studying the temperature dependence of the graphene growth rate, where the growth rate was expressed with the coverage at certain growth time. In the work by Dr. Geohegan et al. (*ACS Nano* 2014, 8, 5657–5669), they used the variation of the graphene coverage over time ($d\theta/dt$) to explore the role of nucleation density in the growth of single-crystal graphene. And in the previous work by our group, the growth time when the full coverage of monolayer graphene is completed has been used to compare the growth rate of ethanol-precursor-based and methane-precursor-based CVD system (*Adv. Mater.* 2017, 29, 1603428).

As the reviewer pointed, it is essential to conduct in-depth analyses for the additional factors that affect graphene growth rate. We have thoroughly examined the thermal and kinetic parameters as well as the electronic information necessary in three fundamental processes (decomposition of carbon sources, adsorption of active carbon species, and graphene growth) to achieve a more precise and focused understanding of graphene growth. We have supplemented a list of the thermal and kinetic parameters (e.g., heat of reaction (ΔE), adsorption energy (E_{ads}), activation energy (E_a), threshold energy barriers, etc.) affecting graphene growth behaviors of interest to the reviewers in **Response Table 1–3**, in addition to the graphing data in the manuscript and supporting information.

(1) Decomposition of carbon sources:

As shown in **Response Table 1**, **Fig. 3a** and revised **Supplementary Fig. 12**, CH_2Cl_2 has a clear tendency to decompose thermally and kinetically more than that of CH_4 due to lower stepwise pyrolysis ΔE , E_a and threshold barriers (E_a (RL), 6.17 eV for CH_4 and 5.59 eV for CH_2Cl_2) (**Response Table 1**). Meanwhile, according to the thermodynamic equilibrium calculations in **Response Fig. 15** (also revised **Supplementary Fig. 13**), CH_2 from CH_2Cl_2 pyrolysis is not only more cleaved than CH_3 from CH_4 pyrolysis, but its concentration is also noticeably higher under

the same carbon supply and temperature. For example, the partial pressure of CH₂ produced by CH₂Cl₂ pyrolysis (1.82×10^{-8} bar) is two orders of magnitude higher than the partial pressure of CH₃ produced by CH₄ pyrolysis (1.37×10^{-6}) at 1373 K, indicating a more abundant carbon species supply from the CH₂Cl₂ carbon source, thus favoring the graphene growth rate enhancement.

Response Table 1 | Thermal and kinetic parameters during stepwise pyrolysis of dichloromethane and methane carbon sources. Note: the ΔE and E_a represent the reaction heat and reaction activation energy of each elementary reaction, respectively. The E_a with the largest value is highlighted in red as the threshold barrier of the whole pyrolysis process.

CH ₄			CH ₂ Cl ₂		
Step	ΔE (eV)	E_a (eV)	Step	ΔE (eV)	E_a (eV)
CH ₄ → CH ₃ + H	6.02	6.02	CH ₂ Cl ₂ → CH ₂ Cl + Cl	2.30	3.60
CH ₃ → CH ₂ + H	6.17	6.17	CH ₂ Cl → CH ₂ + Cl	4.20	5.31
CH ₂ → CH + H	5.59	5.59	CH ₂ → CH + H	5.59	5.59
CH → C + H	5.51	5.51	CH → C + H	5.51	5.51

Response Fig. 15 | Calculated partial pressure of the decomposition products of (a) methane and (b) dichloromethane as a function of temperature. The reaction is assumed under a thermodynamic equilibrium, and $P(\text{CH}_4) = P(\text{CH}_2\text{Cl}_2) = 2.5 \times 10^{-4}$ bar, $P(\text{H}_2) = 2.0 \times 10^{-3}$ bar. The typical graphene growth temperature of ~ 1100 °C (~ 1373 K) is marked with grey line.

(2) Adsorption of active carbon species:

As shown in Fig. 3b, c and Response Table 2, which has been added in the Supporting Information as Supplementary Table 1, the carbon species provided by pyrolysis of the two

carbon sources have different adsorption capacities on the SiO₂ surface. Specifically, the stabilized adsorption of CH₃ provided by CH₄ on the SiO₂ surface usually occurs at the surface Si sites with an adsorption energy (E_{ads}) of -0.60 eV, whereas CH₂ supplied by CH₂Cl₂ adsorbs at the active Si site has a much lower E_{ads} of -1.24 eV (**Response Table 2**), resulting in a better thermodynamic stability. It is also worth noting that the active Cl atoms produced by CH₂Cl₂ pyrolysis have a co-adsorption effect with CH₂ (refer to **Fig. 3b** in the manuscript for a detailed explanation), which can significantly reactivate the inert O sites on the SiO₂ surface (decreasing the E_{ads} of CH₂ on O sites from 1.40 to -0.63 eV), and further facilitate the deposition of CH₂ on the substrate surface, thus accelerating the subsequent nucleation and growth of graphene.

Response Table 2 | Comparison of adsorption energies (E_{ads}) on the SiO₂ surface for the dominant carbon species (CH₃, formed by methane pyrolysis, and CH₂ produced by dichloromethane pyrolysis). The thermodynamic stability of the adsorption increases with decreasing negative E_{ads} .

	CH ₄	CH ₂ Cl ₂
Dominant active carbon species	CH ₃	CH ₂
E_{ads} (eV)	-0.60 (CH ₃ -Si)	1.40 (CH ₂ -O) -1.24 (CH ₂ -Si) -0.63 (CH ₂ &Cl-O) -1.26 (CH ₂ &Cl-Si)

(3) Growth-extension of graphene edge:

The fast growth of graphene also relies on the easy attachment of active carbon species to graphene edges and easy dehydrogenation subsequently happened to change the configuration from sp³ hybridization to sp² hybridization. To simplify this process, we just consider the two basic processes of adding a carbon species to the original graphene growth front (**Response Fig. 16**, which has been added in Supplementary Information as **Supplementary Fig. 16**) and dehydrogenation at the growth edge (**Fig. 3d and Supplementary Fig. 17**) to avoid repetitive calculations. **Response Table 3** reveals the thermal and kinetic details of carbon species attachment and growth edge dehydrogenation processes. For the CH₄ growth route, the E_a for the attachment

of the dominant CH₃ at graphene growth edge is calculated to be 0.34 eV, while the E_a for further dehydrogenation of growing edge assisted by hydrocarbon species is as high as 2.06 eV, thus the edge dehydrogenation process is the rate-limiting step for using CH₄ as a carbon source to grow graphene, and as a result, the threshold barrier E_a (RL) is 2.06 eV. Whereas, for the CH₂Cl₂ growth route, the E_a of the dominant CH₂ attachment at the growth edge is 0.23 eV, while the dehydrogenation process proceeds spontaneously with the assistance of Cl, so the rate-limiting step occurs in the carbon splicing process, and the threshold barrier E_a (RL) is lower at only 0.23 eV. Therefore, the mechanisms of the two carbon sources involved in graphene growth are completely different.

Response Table 3 | Thermal and kinetic parameters of two carbon sources involved in graphene growth. Note, the ΔE and E_a represent the reaction heat and reaction activation energy of each elementary reaction, respectively. The E_a with the largest value is highlighted in red as the threshold barrier of the whole growth process.

		CH ₄	CH ₂ Cl ₂
Dominant active carbon species		CH ₃	CH ₂
Carbon species attachment	ΔE (eV)	2.40	-2.90
	E_a (eV)	-0.34	0.23
Growing edge dehydrogenation	ΔE (eV)	-2.07	-0.63
	E_a (eV)	2.06	0

Response Fig. 16 | Energy profiles and the corresponding structure evolution for CH₃ or CH₂ carbon species attachment at the growing zigzag graphene edge.

We further comprehensively evaluated the specific effects of the aforementioned parameters on the graphene growth rate (ν) according to:

$$\nu = \frac{\Delta L \times C_P}{e^{(E_a(RL) + \Delta E)/k_B T}} \quad (1)$$

where the E_a (RL) and ΔE take the values of the above threshold barriers during graphene growth and the reaction heat for dehydrogenation of graphene growing edge, respectively. $\Delta L = 0.142$ nm is the length of a C–C bond in graphene, and C_P represents the collision rate of active carbon species, which is proportional to the partial pressure of the carbon species in the gas phase (see more details in **Supplementary Fig. 13**). Therefore, the rate ratios of graphene growth by CH₄ and CH₂Cl₂ may be derived by correlating the partial pressures of the dominate active carbon species with the thermal and kinetic parameters of the two carbon sources involved in graphene growth. After substituting the values into Equation 1, it can be obtained that the rate of CH₂Cl₂ involved in

the growth of graphene is about 10^3 times higher than that of CH_4 , which is essentially compatible with the experimental finding shown in **Supplementary Fig. 8**.

In addition, we have added the corresponding descriptions as follows: "*The difference in the rate (v) of graphene growth using CH_4 and CH_2Cl_2 can be comprehensively evaluated by the following equation:*

$$v = \frac{\Delta L \times C_P}{e^{(E_a(RL) + \Delta E)/k_B T}} \quad (1)$$

where the E_a (RL) and ΔE take the values of the threshold barriers during graphene growth and the reaction heat for dehydrogenation of graphene growing edge, respectively. $\Delta L = 0.142$ nm is the length of a C-C bond in graphene, and C_P represents the collision rate of active carbon species⁵¹, which is proportional to the partial pressure of the carbon species in the gas phase (see more details in Supplementary Fig. 13). Therefore, after substituting the values into above equation, it can be obtained that the rate of CH_2Cl_2 involved in the growth of graphene is about 10^3 times higher than that of CH_4 , which is essentially compatible with the experimental finding shown in supplementary Fig. 8" in the revised manuscript (Line 284–294, Page 15).

Comment 2):

The author's argument regarding "the same growth rate" requires additional data for validation. While the author mentions increasing the amount of methane to achieve the same growth rate as dichloromethane, they do not provide specific results regarding the difference in growth rates with respect to the increase in methane. This omission could potentially lead to confusion among readers. Therefore, it is essential to supplement the argument with data that demonstrates how varying the amount of methane affects the growth rate, to substantiate the claim of achieving the same growth rate when using dichloromethane and methane.

Our reply:

Thank you for the constructive comments from the reviewer, particularly about the demonstration of "the same growth rate" for dichloromethane- and methane-grown graphene in **Fig. 2d, e**. According to the reviewer's valuable advice, we have supplemented new data illustrating how varying the amount of methane (actually hydrogen-to-methane (H_2/CH_4)) influences the graphene growth rate (**Response Fig. 17**). In graphene CVD growth system with methane as the carbon precursor, the graphene growth rate can be effectively manipulated by adjusting the H_2/CH_4 . For example, as shown in **Response Fig. 17**, when H_2/CH_4 ratio was modulated from 25 to 0.2, the graphene growth rate can be manipulated from $\sim 6.4 \times 10^{-4}$ to 2 min^{-1} .

Specifically, the growth rate of dichloromethane-grown graphene in **Fig. 2d** was $\sim 2 \text{ min}^{-1}$ (prepared under H_2/CH_2Cl_2 ratio of 8:1, with H_2 flow of 120 sccm and CH_2Cl_2 vapor flow of 15 sccm, CVD growth time ~ 0.5 min). To get the same growth rate as that of dichloromethane-grown graphene, H_2/CH_4 ratio used for methane-grown graphene in **Fig. 2e** was set at 1:5 (with methane flow of 50 sccm and H_2 flow of 10 sccm, CVD growth time ~ 0.5 min). From the data in **Response Fig. 17**, at this H_2/CH_4 ratio, the graphene growth rate was $\sim 2 \text{ min}^{-1}$.

Response Fig. 17 | Growth rates of methane-grown graphene at different H_2/CH_4 ratios, revealing the increased growth rate with H_2/CH_4 ratio reducing.

For a more accurate illustration, we have added **Response Fig. 17** as **Supplementary Fig. 9**,

and we also added the corresponding illustrations in Supplementary information "*In graphene CVD growth system with methane as the carbon precursor, the graphene growth rate can be effectively manipulated by adjusting the H₂/CH₄. For example, as shown in Supplementary Fig. 9, when H₂/CH₄ ratio was modulated from 0.2 to 25, the graphene growth rate can be manipulated from $\sim 6.4 \times 10^{-4}$ to 2 min^{-1} . Specifically, the growth rate of dichloromethane-grown graphene in Fig. 2d was $\sim 2 \text{ min}^{-1}$ (prepared under H₂/CH₂Cl₂ ratio of 8:1, with H₂ flow of 120 sccm and CH₂Cl₂ vapor flow of 15 sccm, CVD growth time $\sim 0.5 \text{ min}$). To get the same growth rate as that of dichloromethane-grown graphene, H₂/CH₄ ratio used for methane-grown graphene in Fig. 2e was set at 1:5 (with methane flow of 50 sccm and H₂ flow of 10 sccm, CVD growth time $\sim 0.5 \text{ min}$), and from the data in Supplementary Fig. 9, at this H₂/CH₄ ratio, the graphene growth rate was also $\sim 2 \text{ min}^{-1}$ ".*

In addition, we have revised the corresponding descriptions follows: "*Figure 2d, e presents the transmission electron microscope (TEM) images of dichloromethane- and methane-grown graphene obtained at the same growth rate ($\sim 2 \text{ min}^{-1}$) (see more details in Supplementary Fig. 9 and Methods section 'Dichloromethane precursor-based CVD growth of GGFF' and 'Methane precursor-based CVD growth of GGFF under consistent growth rate with dichloromethane-based growth')*, which reveal the longer-range crystalline order of dichloromethane-grown graphene in contrast to the amorphous structure of the methane-grown graphene" in the revised manuscript (Line 207–213, Page 11).

Comment 3):

Their graphene-based conductive fiber material, which has the advantage of large area, require application technologies that go beyond those previously reported such as small patch for detecting pulse waves and sound waves. Highly sensitive strain sensors that can be

attached to the skin in a small area have already been reported using various materials (e.g., fibers, conductive polymers, and composites).

Our reply:

Great thanks for the reviewer's positive comment for our GGFF in terms of the advantage of large area, and the constructive advice for developing applications that leverage the advantages of the large area of GGFF.

As highlighted in the manuscript, GGFF demonstrates significant promise as lightweight, flexible sensors characterized by high sensitivity and portability, which is suitable for the applications in the areas of human motion and physiological signal monitoring. Actually, as the reviewer pointed, the advantages of large-area, scalable production of GGFF position it uniquely for addressing the demands of large-size applications, even beyond the sensor applications.

Actually, we are currently devoted to the applications of the large-area GGFF. For example, as we know, the aircraft wing sensors are the essential parts for ensuring the normal work of aircraft. These sensors are anticipated to play a pivotal role in monitoring various parameters critical to an aircraft's wing, such as the strain and deformation, temperature, as well as the structural integrity (*Aerospace 2016, 3, 1; Mechanical Systems and Signal Processing 2020, 136, 106526; Sensors 2019, 19, 55*). The data gathered through these sensors can be swiftly transmitted to both the flight crew and ground control. This capability enables immediate responses to any emerging issues during flight, contributing to enhanced safety and efficiency in aviation operations. GGFF holds promising application potentials in the above scenarios, *i.e.* aircraft wing sensors: (1) as illustrated in **Fig. 4** in the main text, the GGFF-based sensor exhibited high sensitivity for the resistance response, which can ensure the excellent signal detections like the strain and deformation of the target positions; (2) GGFF presented excellent flexibility, which can realize a conformal fit with the objects of different shapes to realize the effective signal acquisitions; (3) The advantage of large

area of GGFF can guarantee the applications on the large-sized aircraft's wings. Therefore, in the fields of aircraft sensors, the large-area GGFF will present the expected application values.

Besides the sensor applications, we are also exploring the applications of large-area GGFF in the fields of electric heating anti/de-icing, electromagnetic shielding, and flexible electronics (*Adv. Mater.* 2022, 34, 2202982; *ACS Nano* 2022, 16, 2577–2584, *Adv. Sci.* 2022, 9, 2105004). For example, the demand for efficient and scalable anti/de-icing solutions has surged in recent years, especially in critical sectors of aviation and renewable energy, for example, the anti/de-icing of aircraft and wind turbine blades. Our GGFF exhibited exceptional electrical heating performances with high heating rate and excellent large-area heating uniformity (for example, it can achieve the heating rate of $\sim 190\text{ }^{\circ}\text{C}\cdot\text{s}^{-1}$ at power density of $\sim 9.3\text{ W}\cdot\text{cm}^{-2}$) (*ACS Nano* 2022, 16, 2577–2584), which provided the new advanced strategy for the electric heating anti/de-icing (*Small Methods* 2022, 6, 2200499). In addition, the excellent structural flexibility of GGFF allows it to conform seamlessly to various surfaces, ensuring comprehensive anti/de-icing protection across expansive areas. Moreover, GGFF has low density of $\sim 2.5\text{ g cm}^{-3}$, which avoids the additional weight gain for the aircraft and wind turbine blades during the large-area practical applications. The large-area coverage, high production capacity, excellent flexibility, lightweight, coupled with the excellent tolerance to harsh environment make GGFF a superior anti/de-icing material for the aircraft and wind turbine blades, which will inject new impetus into the development of aviation and renewable energy areas.

To provide more inspiration to the readers, we have added the corresponding illustrations *"Notably, GGFF's large-area and scalable production capabilities position it uniquely for the large-size applications in various areas. For example, as we know, the aircraft wing sensors are the essential parts for ensuring the normal work of aircraft. These sensors are anticipated to play a pivotal role in monitoring various parameters critical to an aircraft's wing, such as the strain*

and deformation, temperature, as well as the structural integrity^{31,61,62}. GGFF holds promising application potentials in the above scenarios, i.e., aircraft wing sensors. First, according to the analyses in **Fig. 4**, the GGFF-based sensor exhibited high sensitivity for the resistance response. Second, GGFF presented excellent flexibility, which can realize a conformal fit with objects of different shapes to realize the effective signal acquisitions. Therefore, in the fields of aircraft sensors, the large-area GGFF will present the expected application values. Beyond sensors, GGFF also exhibited exceptional electrical heating performances, which made it a promising electric heating material used in the areas of anti/de-icing of large instrument or equipment, such as the aircraft and wind turbine blades. In addition, the excellent structural flexibility of GGFF allows it to conform seamlessly to various surfaces, ensuring comprehensive anti/de-icing protection across expansive areas. Moreover, GGFF has low density of $\sim 2.5 \text{ g cm}^{-3}$, which avoids the additional weight gain for the aircraft and wind turbine blades during the large-area practical applications. The large-area coverage, high production capacity, excellent flexibility, lightweight, as well as the excellent tolerance to harsh environment make GGFF a superior anti/de-icing material for the aircraft and wind turbine blades, which will inject new impetus into the development of aviation and renewable energy areas" in the revised manuscript (Line 353–363, Page 18 and Line 364–372, Page 19).

Comment 4):

In the supplementary Figure 3, it is evident that the highest growth rate is observed when the H₂/ precursor ratio is set at 5 compared to other ratios. This suggests that using a ratio of 5 for the precursor appears to be more effective. However, it appears that the author chose the ratio of 8. It is imperative that the author provides clear justification and explanation for this choice.

Our reply:

We appreciate the reviewer's observation and comment regarding the choice of the H₂/precursor ratio in our research. It should be noticed that in the field of graphene growth, there is usually a trade-off between the growth rate and crystallization quality of obtained graphene. For example, during the growth of large single-crystal graphene with low density of defects, in order to reduce the of defective grain boundaries in CVD graphene, the suppression of graphene nucleation is always required, where a very low amount of carbon precursor supply is usually needed, which, on the other hand, leads to a significantly reduced growth rate of graphene. For example, in the reported work of Prof. Liu et al., the time required for growing micrometer-size graphene grains on SiO₂/Si substrates can be even as long as 3 days (*Adv. Mater.* 2014, 26, 1348–1353). Obviously, there is always a dilemma between the high growth rate and high crystal quality during graphene CVD growth.

The similar dilemma between growth rate and quality can be anticipated in our dichloromethane CVD growth system. On one hand, dichloromethane possesses a lower decomposition energy barrier than methane, thus enabling a large amount of active carbon species produced in CVD growth system to facilitate graphene nucleation and growth. Meanwhile, the introduction of highly electronegative Cl radical can also enhance the capacity of GFF substrate to capture active carbon species, and promote the dehydrogenation and growth of graphene edges. Therefore, the growth processes of graphene on the substrate can be greatly accelerated. On the other hand, excess active carbon specials together with their enhanced adsorption on GFF, in turn leads to the high graphene nucleation density and the limited domain size, which is not conducive to the formation of high crystal quality, and will bring rich defects into the as-grown graphene.

To further illustrate the effects of H₂/CH₂Cl₂ ratio on graphene quality, we supplemented Raman characterization of monolayer graphene grown with H₂/CH₂Cl₂ of 5 (**Response Fig. 18**),

and compared it with graphene grown with $\text{H}_2/\text{CH}_2\text{Cl}_2$ of 8 (orange spectrum in **Fig. 1c**). It can be noticed that under $\text{H}_2/\text{CH}_2\text{Cl}_2$ of 5, the monolayer graphene film could be formed within ~ 20 s with a growth rate of $\sim 3 \text{ min}^{-1}$, faster than that of $\text{H}_2/\text{CH}_2\text{Cl}_2$ of 8 with a growth rate of $\sim 2 \text{ min}^{-1}$. In return, the Raman I_D/I_G ratio of graphene grown with $\text{H}_2/\text{CH}_2\text{Cl}_2$ of 5 is ~ 1.6 , obviously higher than that of graphene grown with $\text{H}_2/\text{CH}_2\text{Cl}_2$ of 8 (~ 1.4), which revealed the lower crystal quality of graphene grown with $\text{H}_2/\text{CH}_2\text{Cl}_2$ of 5. In summary, although the growth condition of $\text{H}_2/\text{CH}_2\text{Cl}_2$ of 5 can achieve a faster growth rate of graphene, it will cause greater sacrifice of graphene crystal quality. In contrast, the growth condition of $\text{H}_2/\text{CH}_2\text{Cl}_2$ of 8 can achieve a better tradeoff between the growth rate and crystal quality, which is the main consideration we chose the growth parameter of $\text{H}_2/\text{CH}_2\text{Cl}_2$ of 8 in our work.

Response Fig. 18 | Comparison of Raman spectra between graphene grown on GFF with $\text{H}_2/\text{CH}_2\text{Cl}_2$ of 8:1 and 5:1 (normalized to G peak intensity). Growth parameters: growth temperature of ~ 1100 °C, H_2 flow of 120 sccm, CH_2Cl_2 flow of 15 sccm for $\text{H}_2/\text{CH}_2\text{Cl}_2$ of 8:1 and CH_2Cl_2 flow of 24 sccm for $\text{H}_2/\text{CH}_2\text{Cl}_2$ of 5:1.

Comment 5):

It would be good to show more details, such as a temperature profile diagram of the CVD process along with the flow rate of the precursors in the experiments performed by the author. (*Journal of Materials Science*, 2020, 55: 545-564., *Science*, 2013, 342.6159: 720-723.)

Our reply:

Great thanks to the reviewer's advice for providing more experimental details in the text, such as the temperature profile diagrams of the CVD process and information about the flow rate of precursors. As the reviewer suggested, we have supplemented the temperature profile diagrams of dichloromethane precursor-based CVD growth of GGFF and methane precursor-based CVD growth of GGFF, which contained the information of flow rates of precursors, growth temperature and other information related to the growth process (**Response Fig. 19**). To make the growth process more visualized, we have added **Response Fig. 19** as **Supplementary Fig. 1** in the revised supplementary information, and **cited the related works of *J. Mater. Sci.* 2020, 55, 545–564 and *Science* 2013, 342, 720–723 in the Reference #63 and #64 in the revised manuscript, respectively**. This additional information will offer readers a more comprehensive view of our experimental setup and conditions.

In addition, we have revised the corresponding descriptions as follows: "*The preparation process of GGFF through the direct CVD graphene growth on GGF is schematically presented in Fig. 1a, where dichloromethane was creatively introduced into the CVD system as the carbon precursor (see more details in the Methods section and Supplementary Fig. 1)*" in the revised manuscript (**Line 107–110, Page 6**).

Response Fig. 19 | Typical CVD graphene growth process in (a) dichloromethane and (b) methane CVD growth system. In a typical procedure, the system was evacuated to a base pressure of <1 Pa and heated to ~ 1100 °C under a H_2 flow of 120 sccm. Subsequently, dichloromethane vapor or methane with desired flow (5–24 sccm) was pumped to the chamber for the graphene growth. Throughout the growth process, the chamber pressure was held at approximately 400–500 Pa depending on the flow of dichloromethane vapor or methane. The growth of graphene lasted for 0.5–10 min for dichloromethane and 8–15 h for methane, followed by natural cooling process to room temperature under a H_2 flow of 20 sccm and Ar flow of 200 sccm.

Comment 6):

Demonstrating the application shown in Figure 4(e) through a live video would be more effective in highlighting to the reader the performance of the fabricated device and demonstrating its higher reliability.

Our reply:

Thank the reviewer for the valuable suggestion about incorporating a live video demonstration of the application presented in Fig. 4e, which presented the response of GGFF sensor attached on a loudspeaker playing a burst of birdsong. According to the reviewer’s advice, we have supplemented the live video presenting the collected $\Delta R/R_0$ of GGFF sensor (in the uploaded attachment). It presented that the detected signals had a synchronous response to the original audio frequency, which intuitively displayed the device's performances. To enhance the overall

presentation of our work, we have added the live video as **Supplementary Video 1** in the revised manuscript.

Comment 7):

Authors should indicate information about bending angles in Supplementary Figure 12

Our reply:

Thank the reviewer for this advice for adding the information about bending angles. We have supplemented the specific information about bending angles in revised **Supplementary Fig. 19** in the revised Supplementary Information, which was also presented below as **Response Fig. 20**.

Response Fig. 20 | $\Delta R/R_0$ of GGFF pressure sensor at different finger bending angles, exhibiting the rapid relative resistance variations with good repeatability and stability.

Comment 8):

A description should be added of which samples were used for analysis (Figure 2 (d), (e) and Figure 4 pressure sensor application).

Our reply:

Thank the reviewer for the suggestion to provide clearer descriptions for the samples used for analyses in specific figures. In our revised manuscript, we have first supplemented the more

detailed descriptions about the experiment method in Method section, and added the corresponding illustrations in the revised manuscript to specify which samples were utilized for analysis in different experiments.

Methods

Dichloromethane precursor-based CVD growth of GGFF

Commercially available GFF of ~0.1 mm thickness (Wuhan Sino Type Optoelectronic Technology CO., LTD) was first annealed under ~500 °C in ambient air for 2 h to remove the coated polymer (same treatment for below). After being carefully cleaned, GFF was placed at the center of a high-temperature furnace with the low-pressure CVD (LPCVD) system. In a typical procedure, the system was evacuated to a base pressure of <1 Pa and heated to 1100 °C under a H₂ flow of 120 sccm. Subsequently, dichloromethane vapor with desired flow (5–24 sccm) was pumped to the chamber for the graphene growth. Throughout the growth process, the chamber pressure was held at approximately 400–500 Pa depending on the flow of dichloromethane vapor. The growth of graphene lasted for 0.5–10 min, followed by natural cooling process to room temperature under a H₂ flow of 20 sccm and Ar flow of 200 sccm. The temperature profile diagrams of the CVD process and information about the flow rate of precursors were shown in Supplementary Fig. 1a^{64,65}. GGFF samples for sheet resistance mapping (Fig. 1f) and TEM characterization (Fig. 2d) were grown for ~10 min and 0.5 min, respectively, with the H₂ flow of 120 sccm and the dichloromethane vapor flow of 15 sccm at ~1100 °C.

Methane precursor-based CVD growth of GGFF under consistent carbon supplies with dichloromethane-based growth

The same GFF substrates and LPCVD method were used in experiments. The system was evacuated to a base pressure of <1 Pa and heated to ~1100 °C under a H₂ flow of 120 sccm. Subsequently, methane gas with desired flow (5–24 sccm) was introduced to the chamber for the

graphene growth. Throughout the growth process, the chamber pressure was held at approximately 400–500 Pa depending on the flow of methane. In each individual comparative experiment, the methane flow remained consistent with the flow of dichloromethane vapor to ensure consistency across the comparative experiments. The growth of graphene lasted for 8–15 h, followed by natural cooling process to room temperature under a H₂ flow of 20 sccm and Ar flow of 200 sccm. The temperature profile diagrams of the CVD process and information about the flow rate of precursors were shown in Supplementary Fig. 1b^{63,64}.

Methane precursor-based CVD growth of GGFF under consistent growth rate with dichloromethane-based growth

To obtain the same growth rate of methane-grown graphene (Fig. 2e) with that of dichloromethane-grown graphene, the H₂/CH₄ ratio was further reduced to 1:5 with the H₂ flow of 10 sccm and the methane flow of 50 sccm. The growth temperature was maintained at ~1100 °C and the CVD growth process lasted for ~0.5 min.

Fabrication of GGFF flexible pressure sensors

GGFF for the pressure sensor fabrication was synthesized with the H₂ flow of 120 sccm and the methane flow of 15 sccm at 1100 °C for ~4 min. Subsequently, copper double-sided tapes were stuck on the both ends of GGFF to serve as the electrodes. Also, copper wires were connected to make an electrical connection. Finally, polypropylene films tightened by a hot plate were used to package the sensor.

Subsequently, for samples in Figure 2 (d), (e) and Figure 4 mentioned by the reviewer, we have added the corresponding illustrations in the revised manuscript to specify which samples were utilized for analyses, which are also listed below.

(1) *Figure 2d, e presents the transmission electron microscope (TEM) images of dichloromethane- and methane-grown graphene obtained at the same growth rate (~2 min⁻¹) (see*

more details in Supplementary Fig. 9 and Methods section ‘Dichloromethane precursor-based CVD growth of GGFF’ and ‘Methane precursor-based CVD growth of GGFF under consistent growth rate with dichloromethane-based growth’), which reveal the longer-range crystalline order of dichloromethane-grown graphene in contrast to the amorphous structure of the methane-grown graphene. (Line 207–213, Page 11)

(2) As shown in Supplementary Fig. 18, a flexible sensor was fabricated based on GGFF encapsulated with the polypropylene films (see more details in Methods section ‘Fabrication of GGFF flexible pressure sensors’). (Line 326–329, Page 17)

In addition, we have also checked the entire text, and added more necessary information for the samples illustrated in the manuscript, which are also listed below.

(1) The preparation process of GGFF through the direct CVD graphene growth on GGF is schematically presented in Fig. 1a, where dichloromethane was creatively introduced into the CVD system as the carbon precursor (see more details in Supplementary Fig. 1a and Methods section ‘Dichloromethane precursor-based CVD growth of GGFF’). (Line 107–110, Page 6)

(2) The sheet resistance mapping on GGFF in Fig. 1f reveals the high uniformity of the electrical conductivity with a mean value of $35.0 \pm 2.3 \Omega \text{ sq}^{-1}$ over the $5 \text{ cm} \times 5 \text{ cm}$ area (see more details in Methods section ‘Dichloromethane precursor-based CVD growth of GGFF’). (Line 155–158, Page 8)

(3) The sheet resistance of GGFF as a function of growth time obtained with dichloromethane and methane precursors was systematically compared in Fig. 2b (carbon supplies introduced into the two CVD systems remained consistent) (see more details in Methods section ‘Dichloromethane precursor-based CVD growth of GGFF’ and ‘Methane precursor-based CVD growth of GGFF under consistent carbon supplies with dichloromethane-based growth’). (Line 180–184, Page 10)

Comment 9):

I recommend that the authors present GGFF fabricated from methane precursors as shown in Figure 1(b).

Our reply:

Thank you for the reviewer's advice for providing GGFF samples fabricated from methane precursors as shown in Fig. 1b, which showed the photographs of large-area bare GFF and GGFFs (with width of ~25 cm) obtained from dichloromethane precursor with the growth time of ~0.5, 2, 5, and 10 min. We have supplemented the photographs of GGFF obtained from methane precursor with different graphene growth times (H_2 flow of 120 sccm and CH_4 flow of 15 sccm at the growth temperature of ~ 1100 °C), as shown in **Response Fig. 21**. It can be seen that graphene was successfully synthesized on the GFF substrates after the growth time of ~ 8 h in methane system (**Fig. 1b**). The contrasts of GGFFs gradually became darker with growth time extending, indicating the increase of graphene layer thickness.

Response Fig. 21 | Photographs of large-area GGFFs obtained with the graphene growth times of ~ 10 min, 8 h, 11 h, and 13 h (growth parameters: growth temperature of ~ 1100 °C, H_2 flow of 120 sccm, CH_4 flow of 15 sccm).

Comment 10):

There is a typo in the TEM image introduction on page 10. Figure 3 should be changed to Figure 2. And there is a spacing error in the second line ‘constructed in’ in the Figure 4 caption.

Our reply:

Thank the reviewer for pointing out the typo in the TEM image introduction and the spacing error in the caption. In our revised manuscript, we made the corresponding corrections, and we have checked the entire text to avoid the similar errors.

(1) Changed "**Figure 3d, e**" to "**Figure 2d, e**" in the TEM image introduction in the revised manuscript (**Line 207, Page 11**).

(2) Corrected the spacing error in the second line of the Fig. 4 caption.

REVIEWER COMMENTS

Reviewer #1 (Remarks to the Author):

The author has addressed some of the concerns raised during the revision; however, reservations persist concerning the technical data and novelty of the manuscript. I cannot recommend the publication in Nature Communications.

The author acknowledged that Prof. Ding et al.'s prior work (Advanced Science 2022, 9, 2200737) in incorporating dichloromethane into the graphene CVD growth system, leading to the authors' work more as an enhancement or optimization, exerting a notable impact on its innovativeness. The Raman spectra fail to distinctly illustrate the discrepancy in graphene quality between CHCl_3 - and CH_2Cl_2 -. Solely based on the Raman spectrum, D/G intensity ratios are elevated in both cases, indicative of suboptimal graphene film quality. Despite extensive evidence from Raman spectroscopy, the persistently appearing high D band suggests inadequate graphene quality. Consequently, the viability of this method, even for industrial applications, may be questionable. The AFM image reveals graphene on Silicon rather than on glass fiber. Moreover, observable contaminations reminiscent of gel or polymer residues are present on the graphene surface. The irregular shape of graphene evident in SEM figures raises concerns about film uniformity, particularly highlighted in Figure 9b.

I suggest transferring the manuscript to another appropriate journal.

Reviewer #2 (Remarks to the Author):

As requested, I have carefully read through the former comments from Reviewer #1, as well as the responses from the authors. Reviewer #1 is an expert in the graphene growth community, and reviewed this manuscript in a very rigorous way. Some of the comments are informative and worthy to be taken into consideration in this manuscript. Overall, I think the authors have satisfactorily answered these comments, although some further discussions are required to reflect Reviewer #1's concerns.

The first disagreement between the authors and Reviewer 1# is whether a prior work (Ding, et al. Advanced Science 2022, 9, 2200737) compromises the novelty of this manuscript (Wang, et al.). After carefully reading Ding's work as well as this manuscript, I think there are substantial differences between these two works. Firstly, for the carbon sources, Ding's work primarily used a mixed carbon precursor (chloroform (CHCl_3) and methanol (CH_3OH)) to showcase the accelerated decomposition of CH_3OH under the assistance of CHCl_3 , while CH_2Cl_2 was just used as a control. Wang's work introduces CH_2Cl_2 as an independent single carbon precursor for graphene growth, presenting a distinct mechanism for achieving ultrafast graphene growth, which is attributed to the low decomposition energy barrier of CH_2Cl_2 , enhanced adsorption of active carbon species through Cl-CH_2 coadsorption, and facilitated H detachment from graphene edges. Especially, Wang et al. proposed the "multispecies-coadsorption strategy", which is a novel concept in graphene CVD growth. Secondly, Ding's work used a plasma-enhanced CVD (PECVD) as the methodology. In the PECVD process, external plasma generator is used to assist the dissociation of carbon precursors and thus it less relies on intrinsic properties of carbon sources. Wang's work used a conventional thermal CVD, where the thermal dissociation of carbon precursors is usually the rate-determining step. Literally, the incorporation of easy-to-dissociate carbon precursors in thermal CVD represents a more important advance than that in PECVD. Lastly, these two works aim at different graphene materials. Ding's work produced vertical graphene as thermal interface material, while Wang's work produced flat graphene as electrically conductive layers on glass fiber fabric, which developed a new composite material, called as graphene glass fiber fabric. Given the differences in carbon precursors, types of CVD process, and obtained graphene materials, I think that both works represent significant progresses in graphene growth, and Ding's work does not compromise the novelty of Wang's work, from my point of view. I would suggest that the authors discuss more on these differences in the discussion part of the manuscript, in avoid of any novelty concerns.

The second disagreement between the authors and Reviewer 1# is whether the present Raman data adequately illustrate the discrepancy in graphene quality between CHCl_3 - and CH_2Cl_2 -derived graphene (Response Fig. 1). The ID/IG value serves as a crucial metric in assessing graphene quality, with a higher ID/IG value indicating poorer graphene quality. The authors showed that the

ID/IG ratio for CHCl₃-grown graphene is higher (~1.8) than that of CH₂Cl₂-grown graphene (~1.4). I think it is safe to conclude that the quality of CH₂Cl₂-derived graphene is better than that of CH₃Cl-derived one. Indeed, in both cases, the D bands are high, denoting that the graphene quality is not "perfect". Considering that the substrate for graphene growth is a noncatalytic glass fiber and no additional catalyst or plasma assistance was used in Wang's work, I think the obtained graphene quality is good among such kinds of research, although there is room for further optimization. For graphene CVD growth, especially on noncatalytic substrates, we would have to consider the tradeoff between "high quality" and "high growth rate", from the economic and application point of view. Wang's work demonstrated the highest growth rate among reported works for graphene grown on noncatalytic substrates while accompanied by a satisfactory quality of graphene. Actually, the graphene growth community is heading beyond just considering the graphene quality. For real-world applications, the production capacity, cost, scalability, etc., stand out as pivotal factors. Wang's work offers a feasible energy-efficient route to solve the issue of high energy consumption caused by the long-time, high-temperature growth process in the mass production of CVD graphene. So I think Wang's work is an important progress and worthy to be published in this journal.

Lastly, Reviewer #1 raised some concerns about the AFM and SEM characterization of the graphene materials. There seems to be some misunderstanding. The AFM images were acquired on graphene transferred onto silicon substrates. Transferring graphene from the growth substrate to silicon for AFM characterization serves as a protocol for accurate graphene thickness measurement. This is a common practice in this community, and the methodology is reliable. The contaminations observed on the graphene surface are attributed to the transfer process, which is usually achieved by a polymer intermediate thus graphene surface is inevitably contaminated. Since the graphene grown on glass fiber is directly utilized in the follow-up application and does not require a transfer process, the concerns introduced by the transfer process do not represent a problem for the applications. To avoid misunderstanding, I would suggest the authors clearly state in the manuscript and the figure caption about how the characterization was conducted – either as-grown graphene on glass fiber substrate or transferred graphene on silicon. Reviewer #1 also raised concerns about film uniformity, based on the irregular shape of graphene observed in a SEM image (Response Fig. 9b). That SEM image is obtained on graphene coverage of ~85%, as marked by the authors in the picture. Actually according to my experience, there is no concrete link between "domain shape" and "film uniformity". Irregular shaped graphene domains can stitch into a uniform film - this is common sense in graphene growth community. I understand Reviewer 1's concern on film uniformity. Raman is one of best ways to show graphene film uniformity, which is widely accepted in this community. I checked again the Raman mapping data in the response letter (Response Fig. 2) – the uniform contrast is evident that the film uniformity is satisfactory. Overall, in my opinion, the concerns from Reviewer #1 are addressable by adding some more discussion on the background and clarification on the data presentation. The novelty and potential impact of Wang's work should warrant its publication in this journal after these clarifications.

Reviewer #3 (Remarks to the Author):

The authors well addressed the reviewers' comments in the revised manuscript and provided valid supplementary materials. To carry out our comments, the author performed significant additional analysis to better describe the equation for graphene growth rate and further illustrated comparison according to precursor through additional supplementary data. I recommend the paper could be published in Nature Communications without further change.

RESPONSE TO REVIEWERS' COMMENTS

Reply to Reviewer #1

The author has addressed some of the concerns raised during the revision; however, reservations persist concerning the technical data and novelty of the manuscript. I cannot recommend the publication in Nature Communications.

Our reply:

We are grateful for the reviewer's constructive feedback, which will undoubtedly help strengthen the quality of our manuscript. We are committed to addressing these issues and providing a revised version that hopefully addressed all your concerns. Below are our point-to-point responses:

Comment 1:

The author acknowledged that Prof. Ding et al.'s prior work (*Advanced Science* 2022, 9, 2200737) in incorporating dichloromethane into the graphene CVD growth system, leading to the authors' work more as an enhancement or optimization, exerting a notable impact on its innovativeness.

Our reply:

We acknowledge the reviewer's reservation regarding the potential impact of Prof. Ding et al.'s study (*Adv. Sci.* 2022, 9, 2200737) on the novelty of our manuscript. However, after careful examination, it is clear that there are significant differences between our work and Prof. Ding et al.'s work, supporting the originality and significance of our contributions.

Firstly, in terms of carbon sources, just as the **Reviewer #2** kindly mentioned, Prof. Ding's research predominantly relied on a mixed precursor system involving trichloromethane (CHCl_3) and methanol (CH_3OH), emphasizing the accelerated decomposition of CH_3OH facilitated by

CHCl₃, with dichloromethane (CH₂Cl₂) merely serving as a comparative control. In contrast, we introduced CH₂Cl₂ as an independent carbon precursor for graphene CVD growth, unveiling a unique mechanism for achieving rapid graphene synthesis. This mechanism is characterized by the low decomposition energy barrier of CH₂Cl₂, remarkably enhanced adsorption of active carbon species through Cl-CH₂ coadsorption, and promoted detachment of hydrogen from growing graphene edges. Notably, we proposed an innovative concept of "multispecies coadsorption", representing a new mechanism in the fields of graphene CVD growth and possibly extended to broader 2D material synthesis. In our work, the interaction between the pyrolyzed products of CH₂Cl₂ and glass fiber glass substrate was comprehensively clarified, where the adsorption energy calculations revealed that the inactive O sites on SiO₂(0001) surface were activated when Cl atoms were adsorbed. The resulting coadsorption of CH₂ and Cl induced an attractive dipole-dipole interaction, significantly reducing CH₂ adsorption energy and increasing its adsorption lifetime. This substantially promoted the nucleation and growth processes of graphene on the glass fiber fabric (GFF) substrate. To the best of our knowledge, our work is the first work applying the concept of "multispecies coadsorption" into graphene CVD growth.

Secondly, Prof. Ding et al.'s work employed a plasma-enhanced CVD (PECVD) method, while our work used the conventional thermal CVD route. In PECVD system, the precursor dissociation together with carbon active species adsorption and diffusion is greatly boosted by coupling the energy of a plasma generator (*ACS Appl. Mater. Interfaces* 2021, 13, 9561–9579; *Adv. Energy Mater.* 2017, 7, 1700678; *Nanoscale*, 2013, 5, 5180), so that graphene can be produced at a relatively low temperature (650 °C in Prof. Ding et al.'s work). In the thermal CVD process, no external energy is introduced. From this point, the intrinsic decomposition properties of the carbon source undoubtedly play a more decisive role in conventional thermal CVD system than that in PECVD system. Therefore, the strategy of carbon source design is particularly pivotal

in conventional thermal CVD system without the plasma assistance. With rational carbon source design, we realized the rapid growth of graphene on nonmetallic noncatalytic insulating substrates by the simultaneous modulations for a series of elementary steps of graphene CVD growth, *i.e.*, decomposition of carbon precursors, adsorption and nucleation of active carbon species, and further growing coalescence of graphene domains. We believe this represents a significant advancement in graphene thermal CVD growth area.

Lastly, different types of graphene materials were fabricated in these two works. Prof. Ding et al.'s work focused on the production of vertical graphene or graphene nanowalls on traditional two-dimensional flat substrate (*i.e.*, silicon wafers), whereas we aimed to fabricate flat graphene films onto fibrous substrate with diameter around 7 μm , culminating in the development of a novel composite material: graphene glass fiber and its fabric. Glass fiber is a widely used engineering material possessing light weight, high strength, flexibility, and high-temperature tolerance. It serves as a critical base material in the fields such as aerospace, wind energy utilization, and architectural engineering (*Fiberglass and Glass Technology: Energy-Friendly Compositions. Springer; 2010*). The selection of glass fiber and its fabric as a substrate underscores another ingenuity of our approach, where graphene can hitch a ride on this traditional engineering material to realize the practical applications. This integration not only expands the potential applications of commercial glass fiber fabric but also facilitates the realization of graphene's potential application at macroscopic scale (*ACS Nano 2024, 18, 4617–4623*).

Considering the disparities in carbon precursors, CVD processes, and resulting graphene materials, it is evident that both studies represent significant advancements in graphene synthesis and applications. Thus, we believe Prof. Ding et al.'s work does not detract from the novelty and importance of our work.

To provide clarity and context for readers, we have supplemented the main distinctions

between our work and the related reported work in the **Discussion** section of the revised manuscript as follows: "*We noted that trichloromethane and dichloromethane were previously introduced into a plasma-enhanced CVD (PECVD) system to promote the decomposition of mixed precursors. However, the implementation of dichloromethane in conventional thermal CVD system represents a significant advancement since the intrinsic decomposition properties of precursors play a much more vital role in the thermal CVD process than that in the PECVD process*". (Page 19–20, Line 384–389 in Main Manuscript)

Comment 2):

The Raman spectra fail to distinctly illustrate the discrepancy in graphene quality between CHCl_3 - and CH_2Cl_2 -. Solely based on the Raman spectrum, D/G intensity ratios are elevated in both cases, indicative of suboptimal graphene film quality. Despite extensive evidence from Raman spectroscopy, the persistently appearing high D band suggests inadequate graphene quality. Consequently, the viability of this method, even for industrial applications, may be questionable.

Our reply:

We appreciate the reviewer's preciseness regarding discrepancy in graphene quality between CHCl_3 - and CH_2Cl_2 -grown graphene, as revealed by the Raman spectra in **Response Fig. 1**. We have made appropriate modification for the presentation of Raman data in **Response Fig. 1** so as to visually distinguish the relative intensity of D and G band for CHCl_3 - and CH_2Cl_2 -grown monolayer graphene (**Revised Response Fig. 1**). The intensity ratios of D and G peaks (I_D/I_G) ratio in Raman spectra of CHCl_3 -grown graphene exhibits a marked elevation (~ 1.8) compared to that of CH_2Cl_2 -grown graphene (~ 1.4), indicating the clear discrepancy in growth quality.

Revised Response Fig. 1 | Comparison of Raman spectra between CHCl_3 (left)- and CH_2Cl_2 (right)-grown graphene (normalized to G peak intensity). Growth parameters: growth temperature of $1100\text{ }^\circ\text{C}$, H_2 flow of 120 sccm , CHCl_3 or CH_2Cl_2 flow of 15 sccm .

The reviewer also raised questions about the graphene quality regarding relatively high I_D/I_G observed in both cases, which we totally understand. In graphene CVD growth community, achieving high-quality graphene at a high growth rate, particularly on noncatalytic nonmetallic substrates, remains a formidable challenge (*Nat. Commun.* 2015, 6, 6499; *Adv. Mater.* 2014, 26, 1348–1353; *Adv. Mater.* 2019, 31, 1803639). Specifically, to improve the crystal quality and reduce the formation of defects and grain boundaries in CVD graphene films, it is essential to suppress the abundant nucleation, which necessitates a minimal supply of carbon precursor. However, this reduction in precursor supply would inevitably result in a significantly decreased growth rate. The delicate balance between maximizing growth rate while sustaining high graphene quality necessitates strategic optimization of precursor supply. Therefore, as **Reviewer #2** pointed out, the trade-off between "high quality" and "high growth rate" for graphene CVD growth on noncatalytic nonmetallic substrates is a recognized issue in graphene growth community. While

achieving the highest growth rate among reports for graphene grown on noncatalytic nonmetallic substrates, our work also enabled a much better graphene quality than the commonly used methane precursor-based CVD growth under the same growth conditions, as revealed in **Fig. 2d, e** and **Supplementary Fig. 10**. Actually, beyond quality considerations, the industrial applicability of graphene hinges on factors such as production capacity and cost, underscoring the multifaceted nature of graphene synthesis. Our approach offers an efficient and cost-effective solution to mitigate the high energy consumption associated with prolonged high-temperature CVD growth process of graphene in the noncatalytic system, thereby addressing critical concerns in mass production of CVD graphene.

We have revised our manuscript to include the above **Discussion** section as follows: "*As is known to all, the trade-off between high quality and high growth rate for graphene CVD growth on noncatalytic nonmetallic substrates is a recognized issue for CVD graphene, which is also the direction we are committed to in the future research. Beyond quality considerations, the industrial applicability of graphene hinges on factors such as production capacity and cost. Our approach offers an efficient and cost-effective solution to mitigate the high energy consumption associated with prolonged high-temperature CVD growth process of graphene in the noncatalytic system, thereby addressing critical concerns in mass production of CVD graphene*". (Page 20, Line 395–402 in Main Manuscript)

Comment 3):

The AFM image reveals graphene on Silicon rather than on glass fiber. Moreover, observable contaminations reminiscent of gel or polymer residues are present on the graphene surface. The irregular shape of graphene evident in SEM figures raises concerns about film uniformity, particularly highlighted in Figure 9b.

Our reply:

We appreciate the reviewer's comment regarding the AFM and SEM images. We first apologize for any confusion caused by the data presentation of the AFM characterization. The AFM images revealing graphene on silicon in **Fig. 1e** and **Response Fig. 6** aimed to measure the thickness of the obtained graphene layers. There are two commonly used approaches to measure the thickness of CVD graphene: (i) directly measuring the height of graphene domain on its growth substrate by AFM without the transfer process, where graphene films have not fully covered the substrate (*Adv. Mater.* **2014**, *26*, 1348–1353); (ii) measuring the height of CVD-grown continuous graphene films after transferring it onto a flat substrate (usually silicon) by AFM (*Nat. Photonics* **2019**, *13*, 754–759). Since graphene in **Fig. 1e** and **Response Fig. 6** is a continuous film, the height difference between graphene films and the underlying glass fiber cannot be detected directly on the growth substrate. Therefore, we transferred graphene layers from the growth substrate onto silicon to realize the thickness measurement by AFM.

Contaminations observed on the graphene surface are attributed to the transfer process from the growth substrate onto silicon, rather than the growth process itself. The wrinkles, cracks, and contamination are inevitably introduced into graphene during the transferring process (*Nano Lett.* **2011**, *12*, 414–419; *ACS Nano* **2011**, *5*, 9144–9153; *Nano Lett.* **2013**, *13*, 1462–1467). The observable contaminations on graphene surface could be polydimethylsiloxane (PDMS) residues since the PDMS stamp was used to assist graphene transfer (*Nature* **2009**, *457*, 706–710). Large-size contaminations have been kept away from the measuring area in **Fig. 1e** and **Response Fig. 6**. Thus, the accuracy of obtained thickness values was barely affected by the contaminations. It is important to note that the transfer process is just for thickness measurement purposes. In practical applications, graphene grown on GFF is directly used in our devices, so the transfer process is not required and has no effects on the follow-up applications. To realize the transfer-free applications of CVD graphene is the significant motivation for us to develop graphene glass fiber fabric (GGFF).

To avoid the confusion, also according to **Reviewer #2's** suggestion, we have clearly stated the substrate information in AFM and TEM characterizations in the main manuscript, figure caption, and supplementary information, and supplemented the details for AFM and TEM characterizations in the **Methods** section of the revised manuscript, as follows:

(1) Figure caption

Fig. 1e caption "*AFM image and height profile along the dashed orange line of the graphene ribbon transferred onto the silicon substrate*". (Page 7, Line 128–129 in main manuscript)

Fig. 2d, e caption "*High-resolution TEM images on TEM grid of dichloromethane- and methane-grown graphene obtained under the same graphene growth rate*". (Page 9, Line 176–177 in main manuscript)

Supplementary Fig. 6 "*AFM characterizations of graphene ribbons (measured after being transferred onto the silicon substrate) with different thicknesses obtained through growth time modulation during graphene CVD growth process*". (Supplementary information)

(2) Results section

"*As presented in Fig. 1e, ~2 nm thickness of the graphene ribbon (measured after being transferred onto the silicon substrate), twice the thickness of the grown graphene layer, is corresponding to 1–2 layers of the CVD-grown graphene*". (Page 8, Line 149–152 in main manuscript)

"*Fig. 2d, e presents the transmission electron microscope (TEM) images on TEM grid of dichloromethane- and methane-grown graphene obtained at the same growth rate*". (Page 11, Line 207–208 in main manuscript)

(3) Methods

"Graphene transfer for AFM and TEM characterizations

The graphene transfer process for AFM and TEM characterizations was carried out under the

assistance of polydimethylsiloxane (PDMS) stamps. First, GGFF was put on a PDMS stamp and flattened with a glass plate. After the adhesion of PDMS, the GGFF/PDMS assembly was immersed in hydrofluoric acid solution (20 wt %) for 8 h to etch the GFF substrate, followed by a repeated rinsing process with deionized water. The resulting graphene/PDMS assembly was then pressed onto the target substrate (silicon for AFM and TEM grid for TEM) at 80 °C for 2 h. Finally, the PDMS stamp was carefully peeled off from the substrate, resulting in the transferred graphene layer on the target substrate". (Page 22–23, Line 443–451 in main manuscript)

Furthermore, the irregular shape of graphene, observed at a coverage stage of ~85% in SEM images (**Response Fig. 9b**), can be attributed to the natural coalescence of graphene domains during their growth process – a typical occurrence in the formation of graphene films. This irregular shape in coalescence process of graphene domains was also observed in numerous previously reported works, including metallic substrates (*Nature* 2021, 596, 519–524; *Science* 2014, 344, 286–289), as well as nonmetallic substrates (*ACS Nano* 2019, 13, 10272–10278). It is important to note that assessing the film uniformity based on SEM images may be misleading due to charge accumulation on the non-conductive glass fiber fabric substrate in the absence of continuous graphene films. As **Reviewer #2** pointed, the more concrete evidence for the uniformity of the as-grown graphene films is provided by the Raman mapping in **Response Fig. 2**, which is widely accepted in the field and offers a more reliable means to characterize graphene films uniformity (*Nat. Mater.* 2022, 21, 740–747; *Science* 2009, 324, 1312–1314; *Adv. Mater.* 2019, 31, 1903615). The intensity mappings of 2D peak in Raman characterization of the fabricated GGFF with different graphene thicknesses have been supplemented in the previous-round revision in the supplementary information as **Supplementary Fig. 3**, and the corresponding illustration "*As revealed in Supplementary Fig. 3, I_{2D} mappings of the fabricated GGFF with varying graphene thicknesses provided the visual representations of high layer uniformity*" has been also added in

the previous-round revised manuscript. (Page 8, Line 141–143 in main manuscript)

Reply to Reviewer #2

As requested, I have carefully read through the former comments from Reviewer #1, as well as the responses from the authors. Reviewer #1 is an expert in the graphene growth community, and reviewed this manuscript in a very rigorous way. Some of the comments are informative and worthy to be taken into consideration in this manuscript. Overall, I think the authors have satisfactorily answered these comments, although some further discussions are required to reflect Reviewer #1's concerns.

The first disagreement between the authors and Reviewer 1# is whether a prior work (Ding, et al. *Advanced Science* 2022, 9, 2200737) compromises the novelty of this manuscript (Wang, et al.). After carefully reading Ding's work as well as this manuscript, I think there are substantial differences between these two works. Firstly, for the carbon sources, Ding's work primarily used a mixed carbon precursor (chloroform (CHCl_3) and methanol (CH_3OH)) to showcase the accelerated decomposition of CH_3OH under the assistance of CHCl_3 , while CH_2Cl_2 was just used as a control. Wang's work introduces CH_2Cl_2 as an independent single carbon precursor for graphene growth, presenting a distinct mechanism for achieving ultrafast graphene growth, which is attributed to the low decomposition energy barrier of CH_2Cl_2 , enhanced adsorption of active carbon species through Cl-CH_2 coadsorption, and facilitated H detachment from graphene edges. Especially, Wang et al. proposed the "multispecies-coadsorption strategy", which is a novel concept in graphene CVD growth. Secondly, Ding's work used a plasma-enhanced CVD (PECVD) as the methodology. In the PECVD process, external plasma generator is used to assist the dissociation of carbon precursors and thus it less relies on intrinsic properties of carbon sources. Wang's work used a conventional thermal CVD, where the thermal dissociation of carbon precursors is usually

the rate-determining step. Literally, the incorporation of easy-to-dissociate carbon precursors in thermal CVD represents a more important advance than that in PECVD. Lastly, these two works aim at different graphene materials. Ding's work produced vertical graphene as thermal interface material, while Wang's work produced flat graphene as electrically conductive layers on glass fiber fabric, which developed a new composite material, called as graphene glass fiber fabric. Given the differences in carbon precursors, types of CVD process, and obtained graphene materials, I think that both works represent significant progresses in graphene growth, and Ding's work does not compromise the novelty of Wang's work, from my point of view. I would suggest that the authors discuss more on these differences in the discussion part of the manuscript, in avoid of any novelty concerns.

The second disagreement between the authors and Reviewer 1# Is whether the present Raman data adequately illustrate the discrepancy in graphene quality between CHCl_3 - and CH_2Cl_2 -derived graphene (Response Fig. 1). The I_D/I_G value serves as a crucial metric in assessing graphene quality, with a higher I_D/I_G value indicating poorer graphene quality. The authors showed that the I_D/I_G ratio for CHCl_3 -grown graphene is higher (~ 1.8) than that of CH_2Cl_2 -grown graphene (~ 1.4). I think it is safe to conclude that the quality of CH_2Cl_2 -derived graphene is better than that of CH_3Cl -derived one. Indeed, in both cases, the D bands are high, denoting that the graphene quality is not "perfect". Considering that the substrate for graphene growth is a noncatalytic glass fiber and no additional catalyst or plasma assistance was used in Wang's work, I think the obtained graphene quality is good among such kinds of research, although there is room for further optimization. For graphene CVD growth, especially on noncatalytic substrates, we would have to consider the tradeoff between "high quality" and "high growth rate", from the economic and application point of view. Wan's work demonstrated the highest growth rate among reported works for graphene

grown on noncatalytic substrates while accompanied by a satisfactory quality of graphene. Actually, the graphene growth community is heading beyond just considering the graphene quality. For real-world applications, the production capacity, cost, scalability, etc., stand out as pivotal factors. Wang's work offers a feasible energy-efficient route to solve the issue of high energy consumption caused by the long-time, high-temperature growth process in the mass production of CVD graphene. So I think Wang's work is an important progress and worthy to be published in this journal.

Lastly, Reviewer #1 raised some concerns about the AFM and SEM characterization of the graphene materials. There seems to be some misunderstanding. The AFM images were acquired on graphene transferred onto silicon substrates. Transferring graphene from the growth substrate to silicon for AFM characterization serves as a protocol for accurate graphene thickness measurement. This is a common practice in this community, and the methodology is reliable. The contaminations observed on the graphene surface are attributed to the transfer process, which is usually achieved by a polymer intermediate thus graphene surface is inevitably contaminated. Since the graphene grown on glass fiber is directly utilized in the follow-up application and does not require a transfer process, the concerns introduced by the transfer process do not represent a problem for the applications. To avoid misunderstanding, I would suggest the authors clearly state in the manuscript and the figure caption about how the characterization was conducted – either as-grown graphene on glass fiber substrate or transferred graphene on silicon. Reviewer #1 also raised concerns about film uniformity, based on the irregular shape of graphene observed in a SEM image (Response Fig. 9b). That SEM image is obtained on graphene coverage of ~85%, as marked by the authors in the picture. Actually according to my experience, there is no concrete link between “domain shape” and “film uniformity”. Irregular shaped graphene domains can

stitch into a uniform film— this is common sense in graphene growth community. I understand Reviewer 1#'s concern on film uniformity. Raman is one of best ways to show graphene film uniformity, which is widely accepted in this community. I checked again the Raman mapping data in the response letter (Response Fig. 2) – the uniform contrast is evident that the film uniformity is satisfactory.

Overall, in my opinion, the concerns from Reviewer #1 are addressable by adding some more discussion on the background and clarification on the data presentation. The novelty and potential impact of Wang's work should warrant its publication in this journal after these clarifications.

Our reply:

We are really grateful for the reviewer's professional and objective evaluation about our work, as well as the constructive feedback, which undoubtedly help strengthen our manuscript. According to the reviewer's advice, we have added more discussions on the background and clarification on the data presentation to clarify the confusion and address the concerns from **Reviewer #1**. We are committed to providing a revised version that meets the standards expected for publication.

According to the reviewer's suggestion about discussing more on the differences and novelty of our work compared with the related reported work, we have added the illustration about the main distinctions of our work in the **Discussion** section of the revised manuscript as follows: "*We noted that trichloromethane and dichloromethane were previously introduced into a plasma-enhanced CVD (PECVD) system to promote the decomposition of mixed precursors. However, the implementation of dichloromethane in conventional thermal CVD system represents a significant advancement since the intrinsic decomposition properties of precursors play a much more vital role in the thermal CVD process than that in the PECVD process*". (Page 19–20, Line 384–389 in main manuscript)

According to the Reviewer's advice to state how the characterization was conducted, we have clearly pointed out the substrate information in AFM and TEM characterizations in the main manuscript, figure caption, and supplementary information, and also supplemented the details for AFM and TEM characterizations in the **Methods** section in the revised manuscript, as follows:

(1) Figure caption

Fig. 1e caption "*AFM image and height profile along the dashed orange line of the graphene ribbon transferred onto the silicon substrate*". (Page 7, Line 128–129 in main manuscript)

Fig. 2d, e caption "*High-resolution TEM images on TEM grid of dichloromethane- and methane-grown graphene obtained under the same graphene growth rate*". (Page 9, Line 176–177 in main manuscript)

Supplementary Fig. 6 "*AFM characterizations of graphene ribbons (measured after being transferred onto the silicon substrate) with different thicknesses obtained through growth time modulation during graphene CVD growth process*". (Supplementary information)

(2) Results section

"As presented in Fig. 1e, ~2 nm thickness of the graphene ribbon (measured after being transferred onto the silicon substrate), twice the thickness of the grown graphene layer, is corresponding to 1–2 layers of the CVD-grown graphene". (Page 8, Line 149–152 in main manuscript)

"Fig. 2d, e presents the transmission electron microscope (TEM) images on TEM grid of dichloromethane- and methane-grown graphene obtained at the same growth rate". (Page 11, Line 207–208 in main manuscript)

(3) Methods

"Graphene transfer for AFM and TEM characterizations

The graphene transfer process for AFM and TEM characterizations was carried out under the

assistance of polydimethylsiloxane (PDMS) stamps. First, GGFF was put on a PDMS stamp and flattened with a glass plate. After the adhesion of PDMS, the GGFF/PDMS assembly was immersed in hydrofluoric acid solution (20 wt %) for 8 h to etch the GFF substrate, followed by a repeated rinsing process with deionized water. The resulting graphene/PDMS assembly was then pressed onto the target substrate (silicon for AFM and TEM grid for TEM) at 80 °C for 2 h. Finally, the PDMS stamp was carefully peeled off from the substrate, resulting in the transferred graphene layer on the target substrate". (Page 22–23, Line 443–451 in main manuscript)

REVIEWERS' COMMENTS

Reviewer #2 (Remarks to the Author):

The authors carefully revised the manuscript and added experiments and detailed explanations to the reviewer's questions. For Reviewer 1, the authors clarified the significant differences between their research and previous work, emphasizing the novelty and impact of their contributions. The authors have also modified the data representation of Raman spectroscopy to visually distinguish the intensity ratio (ID/IG) of D and G peaks, clarify the purpose of AFM measurement, and explain that the process of transferring to silicon is necessary for accurately measuring the thickness of graphene films. For Reviewer 2, the authors added additional discussion on the background and clarified the data presentation to alleviate confusion and issues raised by the reviewers. In addition, the authors have added detailed illustrations in the discussion section of the revised manuscript, emphasizing the unique aspects and progress of their work compared to relevant research.

Based on the author's revised manuscript and their response to the reviewers, I believe that there are no issues with the current manuscript. I recommend accepting and publishing the manuscript on Nature Communications.

RESPONSE TO REVIEWERS' COMMENTS

Reply to Reviewer #2

The authors carefully revised the manuscript and added experiments and detailed explanations to the reviewer's questions. For Reviewer 1, the authors clarified the significant differences between their research and previous work, emphasizing the novelty and impact of their contributions. The authors have also modified the data representation of Raman spectroscopy to visually distinguish the intensity ratio (I_D/I_G) of D and G peaks, clarify the purpose of AFM measurement, and explain that the process of transferring to silicon is necessary for accurately measuring the thickness of graphene films. For Reviewer 2, the authors added additional discussion on the background and clarified the data presentation to alleviate confusion and issues raised by the reviewers. In addition, the authors have added detailed illustrations in the discussion section of the revised manuscript, emphasizing the unique aspects and progress of their work compared to relevant research.

Based on the author's revised manuscript and their response to the reviewers, I believe that there are no issues with the current manuscript. I recommend accepting and publishing the manuscript on Nature Communications.

Our reply:

We are grateful for the reviewer's acknowledgment of the significant differences between our research and previous work, as well as the clarity provided in the representation of Raman spectroscopy data, the purpose of AFM measurements, and the necessity of the transfer process for accurate thickness measurement of graphene films. Moreover, we appreciate the reviewer's positive feedback on the additional discussion provided on the background and the clarification of data presentation to alleviate any confusion.

Based on the reviewer's assessment and recommendation, we are confident that the manuscript is now ready for publication in *Nature Communications*. We sincerely thank the reviewer for the thorough review and support in advancing our research.